# Benchmarking the accuracy of higher order particle methods in geodynamic models of transient flow

Rene Gassmöller[1], Juliane Dannberg[1], Wolfgang Bangerth[2], Elbridge Gerry Puckett[3], and Cedric Thieulot[4]

[1]Department of Geological Sciences, University of Florida
[2]Department of Mathematics, Colorado State University
[3]Department of Mathematics, University of California, Davis
[4]Department of Earth Sciences, Utrecht University

**Correspondence:** Rene Gassmöller (rene.gassmoeller@ufl.edu)

**Abstract.** Numerical models are a powerful tool for investigating the dynamic processes in the interior of the Earth and other planets, but the reliability and predictive power of these discretized models depends on the numerical method, as well as an accurate representation of material properties in space and time. In the specific context of geodynamic models, particle methods have been applied extensively because of their suitability for advection-dominated processes, and have been used in applications such as tracking the composition of solid rock and melt in the Earth's mantle, fluids in lithospheric- and crustal-scale models, light elements in the liquid core, and deformation properties like accumulated finite strain or mineral grain size, along with many applications outside the Earth sciences.

There have been significant benchmarking efforts to measure the accuracy and convergence behavior of particle methods, but these efforts have largely been limited to instantaneous solutions, or time-dependent models without analytical solutions. As a consequence, there is little understanding about the interplay of particle advection errors and errors introduced in the solution of the underlying transient, nonlinear flow equations. To address these limitations, we present two new dynamic benchmarks for transient Stokes flow with analytical solutions that allow us to quantify the accuracy of various advection methods in nonlinear flow. We use these benchmarks to measure the accuracy of our particle algorithm as implemented in the ASPECT geodynamic modeling software against commonly employed field methods and analytical solutions. In particular, we quantify if an algorithm that is higher-order accurate in time will allow for better overall model accuracy and verify that our algorithm reaches its intended optimal convergence rate. We then document that the observed increased accuracy of higher-order algorithms matters for geodynamic applications with an example of modeling small-scale convection underneath an oceanic plate and show that the predicted place and time of onset of small-scale convection depends significantly on the chosen particle advection method.

Descriptions and implementations of our benchmarks are openly available and can be used to verify other advection algorithms. The availability of accurate, scalable, and efficient particle methods as part of the widely used open source code ASPECT will allow geodynamicists to investigate complex time-dependent geodynamic processes, such as elastic deformation, anisotropic fabric development, melt generation and migration, and grain damage.

# 1 Introduction

Numerical models have been a key tool for geoscientists investigating the processes governing plate tectonics and mantle convection. Among the many one could cite, a cross-section of publications include studies of the evolution of mantle heterogeneities over time (e.g. Kellogg and Turcotte, 1990; McNamara and Zhong, 2005; Brandenburg et al., 2008; Gülcher et al., 2021; Jones et al., 2021), the initiation and evolution of plate boundaries (e.g. Tackley, 1998; Bercovici and Ricard, 2014; Baes et al., 2020; Schierjott et al., 2020), the fate of subducted slabs (e.g. Gurnis and Hager, 1988; Billen, 2008; Faccenna et al., 2017; Grima et al., 2020), plume dynamics (e.g. Farnetani and Richards, 1994; Lin and van Keken, 2006; Dannberg and Gassmöller, 2018; Arnould et al., 2020), the dynamics of microplates (e.g. Glerum et al., 2020; Neuharth et al., 2021), and the seismic cycle (e.g. Van Dinther et al., 2013; Van Zelst et al., 2019). Obviously, the usefulness of such dynamic models relies on the accurate approximation of solutions of the equations that describe the processes under consideration. For geodynamic models of the solid Earth, this usually requires solving the Stokes equations governing the flow, and advection(-diffusion) equations governing the transport of thermodynamic properties like temperature or entropy, chemical composition, trace elements, deformation properties like damage, or mineralogical properties like grain size. Established methods for solving the Stokes equations typically treat the fluid as a continuum and are based on the finite-element (e.g. Moresi et al., 2022), finite-difference (e.g. Gerya and Yuen, 2003; Kaus et al., 2016), and finite-volume methods (e.g. Tackley, 2008). In contrast, there is a wide variety of methods for solving the advection equations (van Keken et al., 1997; Puckett et al., 2018), such as particle methods, continuous or discontinuous field methods, marker-chain methods, or volume-of-fluid methods.

Due to their inherent suitability for modeling advection-dominated problems, different variants of particle methods have become popular in the geodynamic modeling community (Weinberg and Schmeling, 1992; van Keken et al., 1997; Tackley and King, 2003; Gerya and Yuen, 2003; McNamara and Zhong, 2004; Gassmöller et al., 2018; Samuel, 2018; Sime et al., 2021). The main advantage of particles in geodynamic applications is that particles advected with the material flow keep their associated material properties; that is, these properties do not diffuse in space as is the case for many field-based methods. It also means that the differential equations for each particle's location are simply ordinary differential equations for which many good solution approaches are available. On the other hand, while errors in particle methods are less apparent than for field methods, they still exist (Tackley and King, 2003; Gassmöller et al., 2019). In particular, previous work has discussed the influence of errors due to interpolating properties from particles to finite element functions representing Stokes discretizations (Thielmann et al., 2014), the influence of the divergence of the computed velocity field on particle distributions (Wang et al., 2015; Pusok et al., 2017; Sime et al., 2021), and the advection of particles over time in spatially variable flow (Gassmöller et al., 2019). However, a source of error in particle advection methods that has, to the best of our knowledge, not been systematically discussed is the error in advecting the particle position in transient, rapidly changing flow (some examples of this can be seen in Gerya and Yuen (2003)). This type of flow is common in geodynamic models of the upper mantle or lithosphere, because employing a visco-plastic or stress-dependent rheology can cause strong nonlinear feedbacks between the current solution and material properties and therefore cause fast changes over time. While the presence of these errors is known, only few studies systematically investigate its influence on geodynamic applications (Trim et al., 2023a, b). This is largely due to the difficulty

of quantifying their influence, as one needs a time-dependent model solution to compare numerical results against, and most currently available benchmarks either rely on instantaneous solutions (Duretz et al., 2011; Zhong, 1996; Zhong et al., 2008; Schmid and Podladchikov, 2003; Kramer et al., 2021), a steady-state solution (Zhong et al., 2008; Gassmöller et al., 2019) or a comparison between several numerical methods without known exact solution (van Keken et al., 1997; Tackley and King, 2003).

In this work, we measure the particle advection error in transient flow using the particle architecture we have developed as part of our work on the Advanced Solver for Planetary Evolution, Convection, and Tectonics (ASPECT; Kronbichler et al., 2012; Heister et al., 2017). We start with a description of the mathematical problem we would like to solve in Section 2, and then present an analysis of the numerical errors that result from the advection of particles in transient flow (Section 3). We develop new benchmarks for transient flow in a box and spherical shell that have known analytical solutions (Section 4), and use these benchmarks to measure the accuracy of the discussed particle advection methods and quantify their influence on the results of the Stokes equations (Section 5). Finally, we illustrate why focusing on the accuracy of particle methods matters for practical geodynamic applications with a model example of small-scale convection developing underneath oceanic lithosphere (Section 6). We conclude in Section 7. Appendix A contains the derivation of the analytical solution for the spherical shell benchmark, and Appendix B contains a more detailed discussion of some benchmark results.

## 2 Governing equations

For the models in this work, we will consider the incompressible Stokes equations using the Boussinesq approximation. They consist of a force balance and a mass continuity equation:

$$-\nabla \cdot (2\eta \dot{\varepsilon}(\boldsymbol{u})) + \nabla p = \rho \boldsymbol{g}, \tag{1}$$

$$\nabla \cdot \boldsymbol{u} = 0, \tag{2}$$

where bold script represents vector quantities, $\boldsymbol{u} = \boldsymbol{u}(\mathbf{x}, t)$ is the velocity, $p = p(\mathbf{x}, t)$ the pressure, $\rho$ the density, $\eta$ the viscosity, and $\boldsymbol{g}$ the gravity. $\dot{\varepsilon}(\boldsymbol{u}) = \frac{1}{2}(\nabla \boldsymbol{u} + \nabla \boldsymbol{u}^T)$ denotes the strain rate tensor. In other words, because inertia is absent, the equations above describe an instantaneous equilibrium.

The models evolve in time because the density and viscosity may depend on time through additional variables $\phi_c = \phi_c(\mathbf{x}, t)$, $c = 1, \ldots, N_c$, where $N_c$ is the number of additional quantities. Examples include the temperature or other thermodynamic quantities, or chemical compositions. These quantities typically satisfy advection-diffusion equations and may be solved through either field- or particle-based approaches. For the purposes of this paper, let us specifically focus on those properties with negligible diffusion (such as chemical compositions or grain sizes); particle methods for applications with non-negligible diffusivity and reactions have been described elsewhere (Gerya and Yuen, 2003; Sime et al., 2022). That is, we consider equations of the form

$$\frac{\partial \phi_c}{\partial t} + \boldsymbol{u} \cdot \nabla \phi_c = H_c, \tag{3}$$

where the $H_c$ are source terms. In Equations (1), (2), and (3), material properties $\eta$, $\rho$ and source terms $H_c$ are then assumed to depend (perhaps non-linearly) on the solution variables $\boldsymbol{u}$, $p$, and $\phi_c$, resulting in a coupled and time-dependent system of equations.

We end this section by noting that while for simplicity we use the *incompressible* Stokes equations, the usefulness of the benchmark models we present below do not rely on this assumption and will be transferable to compressible models. In fact, an accurate solution to the advection equation may matter more in compressible models, because they often contain more coupled terms, such as adiabatic heating (depending on the pressure gradient), the pressure dependence of the density, and additional processes like phase transitions caused by pressure or temperature changes.

## 3   Numerical methodology

Over the past years we have developed a flexible, scalable, and efficient particle architecture (Gassmöller et al., 2018). This work is open-source, and performs well in modern high-performance computing environments. In particular, it supports advanced computational methods such as an adaptively refined, unstructured, and dynamically changing background mesh, parallelization beyond tens of thousands of (CPU) compute cores, storing arbitrary particle properties, and complex nonlinear solvers. The underlying particle infrastructure is application-agnostic and independent of the used discretization for the field-based quantities. Its methods are integrated into the general purpose open source finite element software library deal.II (Arndt et al., 2023) and have been used to model a wide range of geoscientific applications, as well as Navier-Stokes flow, mixing of granular materials, solid-fluid interaction, and laser metal deposition of metallic particles (Popov and Marchevsky, 2022; Arndt et al., 2020; Golshan et al., 2022; El Geitani et al., 2023; Golshan and Blais, 2022; Murer et al., 2022).

We have discussed the numerical methods for most steps of our particle algorithm (Gassmöller et al., 2018, 2019) and Stokes solver (Kronbichler et al., 2012; Heister et al., 2017) in earlier work and refer there for details on the finite-element method, time stepping algorithm, particle generation, advection, and interpolation from particles to grid. Here we will extend this earlier work by developing transient solutions, and focus on how the temporal accuracy of advection methods controls the overall accuracy of a coupled geodynamic model.

### 3.1   Particle advection

In particle methods, the values of fields $\phi_c$ are approximated by advecting particles that carry these field values as "properties". Particles move with the velocity $\mathbf{u}(\mathbf{x},t)$ that results from solving the Stokes equations (1)–(2), and the properties carried by a particle evolve based on the right hand side $H_c$ in equation (3). In other words, the solution of the partial differential equation (3) is approximated by solving an ordinary differential equation (ODE) tracking the position $\mathbf{x}_i = \mathbf{x}_i(t)$ for each particle $i$, and a separate ODE tracking the evolution of the properties carried by the particle:

$$\frac{d}{dt}\mathbf{x}_i(t) = \mathbf{u}(\mathbf{x}_i(t),t), \tag{4}$$

$$\frac{d}{dt}\phi_{c,i}(t) = H_c(\mathbf{x}_i(t),t,\phi_{c,i}(t)). \tag{5}$$

In practice, the exact velocity $\mathbf{u}(\mathbf{x}, t)$ is not available, but only a numerical approximation in space $\mathbf{u}_h(\mathbf{x}, t)$ to $\mathbf{u}(\mathbf{x}, t)$. Furthermore, this approximation is only available at discrete time steps, $\mathbf{u}_h^n(\mathbf{x}) = \mathbf{u}_h(\mathbf{x}, t^n)$ and it needs to be interpolated between time steps if the advection algorithm for integrating equation (4) requires one or more evaluations at intermediate times between $t^n$ and $t^{n+1}$. If we denote this interpolation in time by $\tilde{\mathbf{u}}_h(\mathbf{x}, t)$ where $\tilde{\mathbf{u}}_h(\mathbf{x}, t^n) = \mathbf{u}_h^n(\mathbf{x})$, then the equation being solved is actually

$$\frac{d}{dt}\tilde{\mathbf{x}}_i(t) = \tilde{\mathbf{u}}_h(\tilde{\mathbf{x}}_i(t), t), \tag{6}$$

where $\tilde{\mathbf{x}}_i(t)$ is the exact solution of this equation using the "wrong" velocity field; if $\tilde{\mathbf{u}}_h$ is a good approximation to $\mathbf{u}$, then we hope that $\tilde{\mathbf{x}}(t)$ is a good approximation of $\mathbf{x}(t)$. In practice, however, we can not even compute $\tilde{\mathbf{x}}(t)$, but need to further approximate it via time stepping.

## 3.2 Convergence of particle advection methods

The particle positions contain error contributions from the inexactly known velocity field discussed in the previous subsection, as well as the error introduced by time stepping the ODEs describing particle position and properties. If we denote by $\tilde{\mathbf{x}}_{i,h}(t)$ the numerical approximation to the solution of equation (6), then the error at some time $t$ will typically satisfy a relationship like

$$\|\tilde{\mathbf{x}}_{i,h}(t) - \tilde{\mathbf{x}}_i(t)\| \leq C(t)\Delta t_{\mathrm{p}}^q, \tag{7}$$

where $\Delta t_{\mathrm{p}}$ is the time step used by the ODE solver, which is often an integer fraction of the time step $\Delta t_{\mathbf{u}}$ used to advance the velocity field $\mathbf{u}$. In our application we will choose $\Delta t_{\mathrm{p}} = \Delta t_{\mathbf{u}}$. $q$ is the convergence order of the method, and $C(t)$ is a (generally unknown) constant that depends on the end time $t$ at which one compares the solutions as well as on $\tilde{\mathbf{u}}$. We want to compare this computed solution against exact trajectories using the exact velocity as in equation (4), and then assess the error as $\|\mathbf{x}_i(t) - \tilde{\mathbf{x}}_{i,h}(t)\|$. This quantity will, in the best case, only satisfy an estimate of the form

$$\|\tilde{\mathbf{x}}_{i,h}(t) - \mathbf{x}_i(t)\| = \|(\tilde{\mathbf{x}}_{i,h}(t) - \tilde{\mathbf{x}}_i(t)) + (\tilde{\mathbf{x}}_i(t) - \mathbf{x}_i(t))\|$$
$$\leq \|\tilde{\mathbf{x}}_{i,h}(t) - \tilde{\mathbf{x}}_i(t)\| + \|\tilde{\mathbf{x}}_i(t) - \mathbf{x}_i(t)\|$$
$$\leq C_1(t)\Delta t_{\mathrm{p}}^q + C_2(t)\|\mathbf{u} - \mathbf{u}_h\| + C_3(t)\|\mathbf{u}_h - \tilde{\mathbf{u}}_h\|,$$

with appropriately chosen norms for the second and third term, which represent how accurately the flow field is discretized in space and time. All of these terms can converge to zero at different rates with the mesh size $h$ and the time step size $\Delta t$; as a consequence, each of these terms may be the limiting factor for the overall accuracy of the ODE integrator.

## 3.3 Common particle integrators

Given these considerations, and given that ODE integrators require the expensive step of evaluating the velocity field $\tilde{\mathbf{u}}_h$ at arbitrary points in time and space, choosing a simpler, less accurate scheme can significantly reduce the computation time. In

our work, we have implemented the Forward Euler, Runge–Kutta 2 (RK2) and Runge–Kutta 4 (RK4) schemes (Hairer and Wanner, 1991; Gassmöller et al., 2018), although other methods are available and have been used in geodynamic applications (e.g. Heun's method in Zhong and Hager (2003) and Sime et al. (2021); Runge–Kutta schemes with additional predictor-corrector steps in Weinberg and Schmeling (1992); implicit Euler and BDF2 methods in Furuichi and May (2015); or Adams Bashforth methods in Adamuszek et al. (2016)). We will briefly discuss our selected methods below and will limit ourselves to

a discussion of two different variants of the RK2 integrator, which is sufficient to support our conclusions.

      For simplicity, we will omit the particle index $i$ from formulas in the remainder of this section and will assume that the ODE and PDE time steps $\Delta t_p, \Delta t_{\mathbf{u}}$ are equal. We will therefore simply denote them as $\Delta t$. This is often the case in practice because the velocity field is typically computed with a method that requires a Courant-Friedrichs-Lewy (CFL) number around or smaller than one, implying that particles move no more than by one cell diameter per (PDE) time step. In such cases, even

explicit time integrators for particle trajectories can be used without leading to instabilities, and all of the methods below fall in this category. The formulas in the remainder of this section are, however, obvious to generalize to cases where $\Delta t_p < \Delta t_{\mathbf{u}}$. We will also assume that we have already solved the velocity field up to time $t^{n+1}$ and are now updating particle locations from $\mathbf{x}^n$ to $\mathbf{x}^{n+1}$. In cases where one wants to solve for particle locations *before* updating the velocity field, $\tilde{\mathbf{u}}_h$ can be extrapolated beyond $t^n$ from previous time steps, or particle advection and velocity computation could be iterated in a nonlinear solver

scheme. Because of this choice, the number of Stokes solutions which have to be computed is independent of the choice of particle advection scheme.

      In the following, let us briefly describe some of the common time stepping algorithms, including those we use in this work.

     1. Forward Euler (FE): The simplest method often used is the forward Euler scheme,

$$\tilde{\mathbf{x}}^{n+1} = \tilde{\mathbf{x}}^n + \Delta t \, \tilde{\mathbf{u}}_h(t^n, \tilde{\mathbf{x}}^n).$$

It is only of first order (that is, the exponent in equation (7) is $q = 1$), but cheap to evaluate because it only requires evaluating the velocity solution at an already-computed time point.

     2. Runge–Kutta second order (RK2): Accuracy and stability can be improved by using a second order Runge–Kutta scheme (that is, $q = 2$ in equation (7)). Among the many second-order Runge-Kutta methods, we specifically choose what is commonly referred to as the *(explicit) midpoint method* in which the new particle position is computed as

$$\mathbf{k}_1 = \frac{\Delta t}{2} \tilde{\mathbf{u}}_h(t^n, \tilde{\mathbf{x}}^n),$$

$$\tilde{\mathbf{x}}^{n+1} = \mathbf{x}^n + \Delta t \, \tilde{\mathbf{u}}_h \left( t^n + \frac{\Delta t}{2}, \tilde{\mathbf{x}}^n + \frac{\mathbf{k}_1}{2} \right).$$

      This method requires evaluating the computed velocity at two different locations and two different points in time, including a time point intermediate between (velocity) time steps. Note that there are other RK2 methods, such as Ralston's method, which reduce the theoretical truncation error of the method while maintaining the order of convergence. How-

ever, in our benchmarks the difference in error is small, and the midpoint method allows us to reduce the memory requirement of the algorithm.

3. Runge–Kutta second-order space, first-order time (RK2FOT): In practice many geodynamic modeling packages only store a single velocity solution at a time, which prevents the interpolation of the velocity field at $t^n + \frac{\Delta t}{2}$ used in RK2 from adjacent solutions at $t^n$ and $t^{n+1}$. However, reasonable accuracy can still often be achieved when ignoring the time dependence of the velocity (Gerya and Yuen, 2003; McNamara and Zhong, 2004). We here implement such an advection scheme as a modification to RK2, in which the new particle position is computed as

$$\mathbf{k}_1 = \frac{\Delta t}{2} \tilde{\mathbf{u}}_h(t^n, \tilde{\mathbf{x}}^n),$$

$$\tilde{\mathbf{x}}^{n+1} = \tilde{\mathbf{x}}^n + \Delta t\, \tilde{\mathbf{u}}_h \left( t^n, \tilde{\mathbf{x}}^n + \frac{\mathbf{k}_1}{2} \right).$$

Note how, compared to the RK2 scheme, the velocity is evaluated at the (wrong) time $t^n$ instead of $t^n + \frac{\Delta t}{2}$, but at the correct location $\tilde{\mathbf{x}}^n + \frac{\mathbf{k}_1}{2}$. RK2FOT still requires the evaluation of the velocity at two different points in space, but only a single point in time.

4. Runge–Kutta fourth order (RK4): A further improvement of particle advection can be achieved by a fourth order Runge–Kutta scheme. We choose the most commonly used scheme that computes the new position as

$$\mathbf{k}_1 = \Delta t\, \tilde{\mathbf{u}}_h \left( t^n, \tilde{\mathbf{x}}^n \right),$$

$$\mathbf{k}_2 = \frac{\Delta t}{2} \tilde{\mathbf{u}}_h \left( t^n + \frac{\Delta t}{2}, \tilde{\mathbf{x}}^n + \frac{\mathbf{k}_1}{2} \right),$$

$$\mathbf{k}_3 = \frac{\Delta t}{2} \tilde{\mathbf{u}}_h \left( t^n + \frac{\Delta t}{2}, \tilde{\mathbf{x}}^n + \frac{\mathbf{k}_2}{2} \right),$$

$$\mathbf{k}_4 = \Delta t\, \tilde{\mathbf{u}}_h \left( t^{n+1}, \tilde{\mathbf{x}}^n + \mathbf{k}_3 \right),$$

$$\tilde{\mathbf{x}}^{n+1} = \tilde{\mathbf{x}}^n + \frac{1}{6}\mathbf{k}_1 + \frac{1}{3}\mathbf{k}_2 + \frac{1}{3}\mathbf{k}_3 + \frac{1}{6}\mathbf{k}_4.$$

RK4 requires the evaluation of the velocity at four points in space and three points in time.

The primary expense in all of the methods above is the evaluation of the velocity field $\mathbf{u}_h^n$ and $\mathbf{u}_h^{n+1}$ at arbitrary positions $\mathbf{x}$. Given that the velocity fields $\mathbf{u}_h$ we consider here are often finite element fields defined with shape functions whose values are determined by mapping a reference cell $\hat{K}$ to each cell $K$ using a transformation $\mathbf{x} = \Phi_K(\hat{\mathbf{x}})$, the evaluation at arbitrary points requires the inversion of $\Phi_K$, which is an expensive operation for nonlinear mappings such as those used in deformed or curved geometries.

## 3.4 Particle integrators used in the benchmarks

Based on our earlier work measuring the convergence properties of the integrators described above in analytically known flow (Gassmöller et al., 2018, supp. information) we expect FE and RK2FOT to converge with first order (in $\Delta t$) in time variable flow, while RK2 and RK4 are expected to converge with second order in time. RK2FOT is limited from reaching the potential of RK2 by the use of only a single velocity solution in time, and RK4 in our specific implementation (though not in general)

is limited by only storing two velocity solutions, which only allows for a linear extrapolation from $t^n$ to $t^n + \frac{\Delta t}{2}$ and $t^{n+1}$. Therefore, while there are valid reasons to choose either FE or RK4, we will limit our benchmark results to RK2 and RK2FOT, because we expect them to illustrate the significant difference between algorithms that are first- or second-order accurate in time.

## 4   Deriving transient benchmark solutions

For our benchmarks we want to reduce the coupling between equations (1)–(2) and (3) to a minimum in order to precisely measure the influence of exactly one coupled property. This step also simplifies the construction of the benchmarks. Therefore, we focus on problems with constant viscosity ($\eta = 1$), and no source terms ($H_c = 0$). The advected material property $\phi_c$ we consider here is the density $\rho$. The final set of equations for our benchmarks will therefore be:

$$-\nabla \cdot (2\dot{\varepsilon}(\boldsymbol{u})) + \nabla p = \rho \boldsymbol{g}, \tag{8}$$

$$\nabla \cdot \boldsymbol{u} = 0, \tag{9}$$

$$\frac{\partial \rho}{\partial t} + \boldsymbol{u} \cdot \nabla \rho = 0. \tag{10}$$

The set of benchmarks we will consider is an extension of previously published benchmarks (Gassmöller et al., 2019). In order to explain the extensions to transient flow, we will here briefly revisit our approach to derive steady flow fields. In our earlier work we have considered incompressible flow fields $\boldsymbol{u}$ that were derived based on a known and time-independent stream function $\Psi$. Under the assumption that viscosity is known and constant, and that boundary conditions are chosen to match the desired solution, this allowed us to compute right-hand side terms to equation (8) that satisfied the set of equations and therefore created an analytical benchmark for the whole system of equations. However, this only guarantees a consistent solution for the distribution of the density $\rho$ at the current instant in time. It is therefore only an *instantaneous* benchmark solution. In order to create a steady-state flow field — defined as a velocity field $\boldsymbol{u}$ that does not change over time — the right-hand side driving force needs to stay constant over time. In other words, the advected property $\rho$ needs to be chosen in such a way that when it is advected with the flow field $\boldsymbol{u}$, the right-hand side $\rho \boldsymbol{g}$ does not change over time. In order to find such a density distribution, we can make use of the definition of the streamline, which are lines of constant $\Psi$. If $\Psi$ is independent of time, any property advected with the flow will be advected along the streamlines. Thus, if $\rho$ is constant along the streamlines, the right-hand-side $\rho \boldsymbol{g}$ will not change even if $\rho$ is advected with the flow. Choosing $\rho = \Psi$ is therefore an easy approach to guarantee a steady-state flow field.

The benchmarks below extend these steady-state models with a nonlinear time-dependence, which will test how much error the chosen advection scheme accumulates over time when the velocity changes. In order to derive such solutions, we make use of the fact that we can superimpose two independent flow fields. In addition to a steady flow based on a stream function $\Psi$ we add a time-dependent velocity that has two special properties. First, we ensure that this second flow field is purely forced by the boundary conditions instead of internal density forces. This choice opens up control over the exact velocity over time. It also implies that we do not have to modify the density distribution to add this second flow field, i.e., the density is still a

function of the steady-state stream function $\Psi$. Second, we will choose the time-variable flow field to be in the nullspace of the Stokes operator, e.g., solid-body rotational flow in a spherical shell and translational flow in a box geometry. This ensures that the resulting modification only affects the velocity solution (but not the pressure), and can be interpreted as a (time-dependent) coordinate transformation of a steady flow. We will consider one case in a two-dimensional spherical shell, and one case in a two-dimensional box geometry and will discuss the specific flow fields in the following subsections.

## 4.1 A benchmark for a 2D spherical shell

As described, we start from an instantaneous solution for Stokes flow in a spherical shell and add a time-dependent rotational flow that is enforced using the boundary conditions. A detailed derivation of the benchmark solution is given in Appendix A. It is important to note that while the benchmark is derived in polar coordinates, it is implemented in Cartesian coordinates in our code. Our final benchmark solution consists of a number of convection cells that rotate around the center of a spherical shell and is described by

$$v_r(r,\theta,t) = g(r)k\sin(k(\theta - \tau(t))), \tag{11}$$

$$v_\theta(r,\theta,t) = f(r)\cos(k(\theta - \tau(t))) + r\omega(t), \tag{12}$$

$$p(r,\theta,t) = kh(r)\sin(k(\theta - \tau(t))), \tag{13}$$

$$\rho(r,\theta,t) = -\left(\frac{A}{2}r^2 + B\ln(r) - 1\right)\cos(k(\theta - \tau(t))), \tag{14}$$

$$g_r(r,\theta,t) = m(r)k\frac{\sin(k(\theta - \tau(t)))}{\rho(r,\theta,t)}, \tag{15}$$

$$g_\theta = 0. \tag{16}$$

The constants $A, B$ and functions $g(r), f(r), h(r), m(r)$ are listed in Appendix A. The parameter $k$ controls the number of upwellings and downwellings in the model and is chosen as $k = 4$ for this study. The parameter $\omega(t)$ represents the time-dependent solid body rotation and is chosen as $\omega(t) = e^t$. $\tau(t)$ is a phase shift caused by the solid body rotation, and is computed as $\tau(t) = \int_0^t \omega(s)\,ds$. The spherical shell has an inner radius of $R_1 = 1$ and an outer radius of $R_2 = 2$. The setup of the benchmark and a snapshot of the solution is shown in Fig. 1.

We note that this solution can be interpreted as consistent with a stream function that is variable in time, with a flow field that conveniently advects the density in such a way as to satisfy our Stokes solution at the current point in time. We also note that this solution effectively consists of two parts, a density-driven internal convection in small convection cells, and a forced and analytically known rotational flow of the whole model.

## 4.2 A benchmark for a 2D box geometry

Our solution for the box benchmark is analogous to the spherical shell case, but we can build directly on our earlier model setup of (Gassmöller et al., 2018). We add a solid body translation, and with periodic boundary conditions at both side boundaries this allows us to define a known, transient solution to the incompressible Stokes equations, which is described by

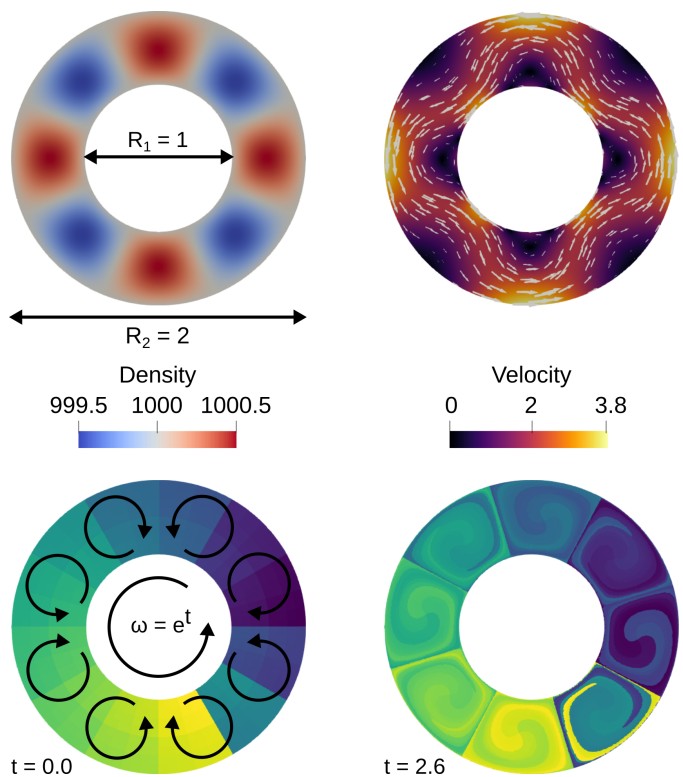

**Figure 1.** *Solution of a transient spherical shell benchmark. Top left: The density field of the benchmark at $t = 0$. Top right: Velocity solution at $t = 0$. Bottom row: Initial ($t = 0$) and later ($t = 2.6$) particle distributions after almost two full rotations of the model. Particles are colored based on a unique index given to each particle at the beginning.*

$$v_x(x, y, t) = \sin(\pi(x - \tau(t))) \cos(\pi y) + \omega(t), \tag{17}$$

$$v_y(x, y, t) = -\cos(\pi(x - \tau(t))) \sin(\pi y), \tag{18}$$

$$p(x, y, t) = 2\pi \cos(\pi(x - \tau(t))) \cos(\pi y), \tag{19}$$

$$\rho(x, y, t) = \sin(\pi(x - \tau(t))) \sin(\pi y), \tag{20}$$

$$g_x(x, y, t) = 0, \tag{21}$$

$$g_y(x, y, t) = -4\pi^2 \frac{\cos(\pi(x - \tau(t))) \sin(\pi y)}{\rho(x, y, t)}. \tag{22}$$

As for the spherical case described above, we will use a nonlinear choice for $\omega(t)$, namely $\omega(t) = e^t$, and the phase shift $\tau$ is computed as before. We choose a box with a width of $w = 2$ and height of $h = 1$. The solution of the benchmark is shown in Fig. 2. The translation of the solution as well as the periodic boundary conditions also represent the main difference between

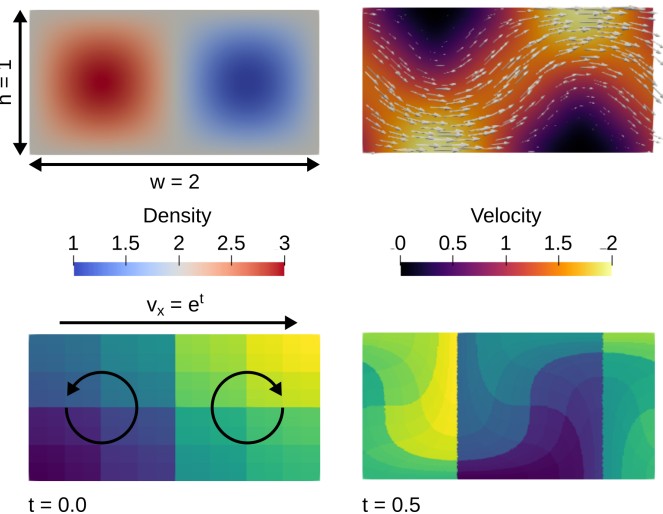

**Figure 2.** *Solution of a transient box benchmark with known analytic solution. Top left: The density field of the benchmark. Top right: Velocity solution. Bottom row: Particle distributions at model start ($t = 0$) and at $t = 0.5$. Particles are colored as in Fig. 1.*

our benchmark solution and the one presented in (Trim et al., 2023a), which uses a steady-state flow with a time-dependent velocity amplitude.

### 4.3 How we use these benchmarks in our particle advection algorithms

Adding time dependence to the benchmarks modifies the numerical solution and the accumulated error in distinct ways, depending on which advection method we choose. Here we will consider five cases:

(1) We obtain a computed solution by using the exact density $\rho(x, y, t)$ defined in equations (14) and (20). This solution will act as our baseline benchmark, illustrating the optimal convergence rate for the Stokes solver we used.

(2) We use the (interpolated) exact density as an initial condition for the density advection equation (10), whose solution we then approximate using discontinuous, piecewise quadratic ($DGQ_2$) finite elements with a penalty method as described in He et al. (2016).

(3) As (2), but we use continuous, piecewise quadratic ($Q_2$) finite elements and an entropy viscosity stabilization technique (Guermond et al., 2011; Kronbichler et al., 2012). This is the default choice in ASPECT. In both (2) and (3) we use a backward differentiation formula (BDF2) time-stepping algorithm that is second-order accurate in time to solve the advection equation (10).

(4) We use the exact density as the initial condition for particles whose position we advect using a second-order accurate Runge-Kutta (RK2) algorithm. Where we need the density for the solution of the Stokes equations, we interpolate

properties from particles onto a $DGQ_2$ discontinuous finite element field and evaluate that at quadrature points as necessary.

(5) As (4), but we use RK2FOT as described in Section 3.3.

In order to limit ourselves to examining *the accuracy in time* of these five benchmark series, we will only consider a single combination of Stokes finite-element and particle interpolation algorithm in this manuscript. We will use a $Q_2 \times Q_1$ Stokes element (continuous, piecewise quadratic velocity, and continuous, piecewise linear pressure), and a linear least-squares particle interpolation algorithm with initially 64 particles per cell. We have described the influence of these choices in earlier work (Gassmöller et al., 2019).

Because the number of particles in a cell can change during the model run, we enforce a minimum of 12 particles per cell, which guarantees that the linear least-squares interpolation algorithm is always sufficiently constrained. We do not limit the maximum number of particles per cell in these models. In practice, the presented benchmarks never require the addition of particles, and therefore the number of particles stays constant (for the box) or decreases by less than $0.01\%$ (for the annulus, caused by integration error close to the boundary, and then leading to the loss of particles across the boundary). The two tested integration schemes do not show a significant difference in particle loss in the annulus geometry, even though we have observed such differences between Runge-Kutta algorithms and Forward Euler integrators in our earlier work (Gassmöller et al., 2018).

## 5 Numerical evaluation of particle schemes

In the following subsections, let us use the benchmarks derived above for the numerical evaluation of particle schemes.

### 5.1 Spherical shell benchmark

Fig. 3 presents the $L_2$-error norm of velocity, pressure, and density for the spherical annulus benchmark at a fixed time as a function of mesh resolution (left column), and at a fixed resolution as a function of time (right column).

The left column illustrates that all advection methods but RK2FOT reach second order convergence for the density with increasing resolution (bottom left panel). As expected RK2FOT is limited by the available time information and only reaches first order convergence. An additional detail is that the field methods ($Q_2$ and $DGQ_2$) have a larger error constant than the particle method (RK2), even for the same convergence rate. We will revisit the source of this error constant when discussing the error accumulation over time. Starting at moderate resolutions (around $h = \frac{1}{16}$) the RK2FOT model only reaches a first order convergence rate in velocity, while $Q_2$ and $DGQ_2$ reach second-order. This result is important, because it illustrates that particles do not uniformly generate smaller errors than field methods, but can indeed generate larger errors if their advection method is too simple and therefore inaccurate. The RK2 method maintains a third-order convergence rate in this metric up to very fine resolutions, which is surprising as the expected convergence order for RK2 is second order. We refer to Section 5.2 for a discussion on how RK2 may reach superconvergence for the resolutions shown in this benchmark.

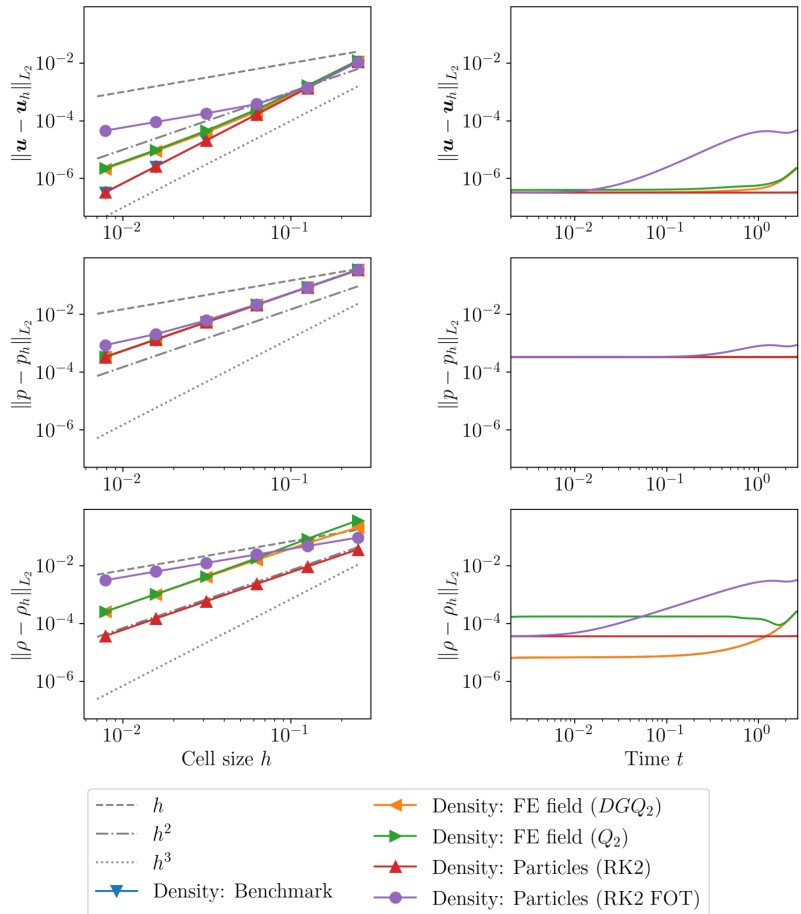

**Figure 3.** *Transient spherical annulus benchmark. Left: $L_2$ error norms of velocity (top row), pressure (middle row), and density (bottom row) for different cell sizes $h$ at time $t = \ln(1 + 4\pi) \approx 2.6075$. Different colors and marker styles show different advection methods; gray lines show ideal first, second, and third order convergence. Note that the line for the exact (benchmark) density overlaps with the RK2 line. Right: $L_2$ error norms of velocity (top row), pressure (middle row), and density (bottom row) over time for resolution $h = 1/128$. Colors as in left column, and the exact benchmark density line is hidden behind the RK2 case. For more details on the distinction between the RK2 case and the benchmark density case see the appendix.*

The analysis of error evolution over time (the right column of Fig. 3) illustrates further differences between field and particle methods. Velocity and pressure errors reveal that RK2FOT accumulates the largest errors over time as expected, followed by $Q_2$ and $DGQ_2$. RK2 accumulates the smallest errors. However, the density error norm shows distinct differences between the methods. Both RK2 and RK2FOT start with the same error value, but while the RK2 error remains near-constant over the evolution of the model (increasing by less than 2%), the error of RK2FOT increases by almost two orders of magnitude. The rate of increase in the RK2FOT scheme changes towards the end of the model run. We show in Appendix B that this slowdown is related to the periodicity of our benchmark solution. The field methods $Q_2$ and $DGQ_2$ behave distinctly different. $DGQ_2$ starts at a much smaller error value than all other methods, but accumulates significant errors towards the end of the model run. $Q_2$ already starts at a significantly larger error value than all other methods. This is likely related to the fact that the used entropy viscosity method falls back to a first-order stabilization scheme for the first time step, which introduces a large amount of numerical diffusion at the model start (and only then). The overall shape of these curves is due to properties of the exact solution, not the method used, but is not of interest to us here.

Summarizing these findings, low-order particle methods show larger errors than the tested field methods, while higher-order particle methods outperform the field methods in our benchmark both with increasing resolution and with increasing model time. Therefore, whenever the other error sources of the solution are sufficiently small (i.e., if the Stokes element and time-stepping scheme allow for higher order accuracy) a higher order particle scheme can significantly improve accuracy.

**5.2  Box benchmark**

The box benchmark results follow a similar pattern for the dependence of errors on the methods used, see Fig. 4. First, the solution using the analytical density solution produces a third-order convergence in velocity and second-order in pressure, which proves that the Stokes elements reach their optimal convergence order when given accurate density distributions. Second, and confirming theoretical predictions, the RK2 first-order time (RK2FOT) advection method creates a first order accurate approximation for the density, which also generates a first-order accurate pressure and velocity solution, therefore significantly limiting the potential accuracy of the Stokes elements. All other advection methods reach second-order convergence as predicted by their derivations, however with significant differences in the absolute error norm. For all solution variables, particles advected using a full RK2 scheme reach about a one order of magnitude lower error norm at the end time than the $Q_2$ and $DGQ_2$ finite-element methods, a value that depends on the chosen end time (compare right column of Fig. 4 and next paragraph). One feature to note is that the velocity error of the RK2 particle advection method starts with a third-order convergence rate at low resolutions and transitions to a second-order convergence rate for higher resolutions (top left panel of Fig. 4, red line with triangles). We assume that this transition is caused by a shift in the dominant error source: At large $h$ the spatial error in the solution dominates, which is consistent with the observation that the particle solution (red line, triangles in bottom left panel of Fig. 4) is very close to the analytical density solution (blue line, circles) and converges with third order. At smaller $h$ the spatial error reduces significantly, leaving the time error, which converges at second-order, as the remaining dominant source of velocity error. We observed such a transition in the dominant error source already in our earlier work (Gassmöller et al., 2019).

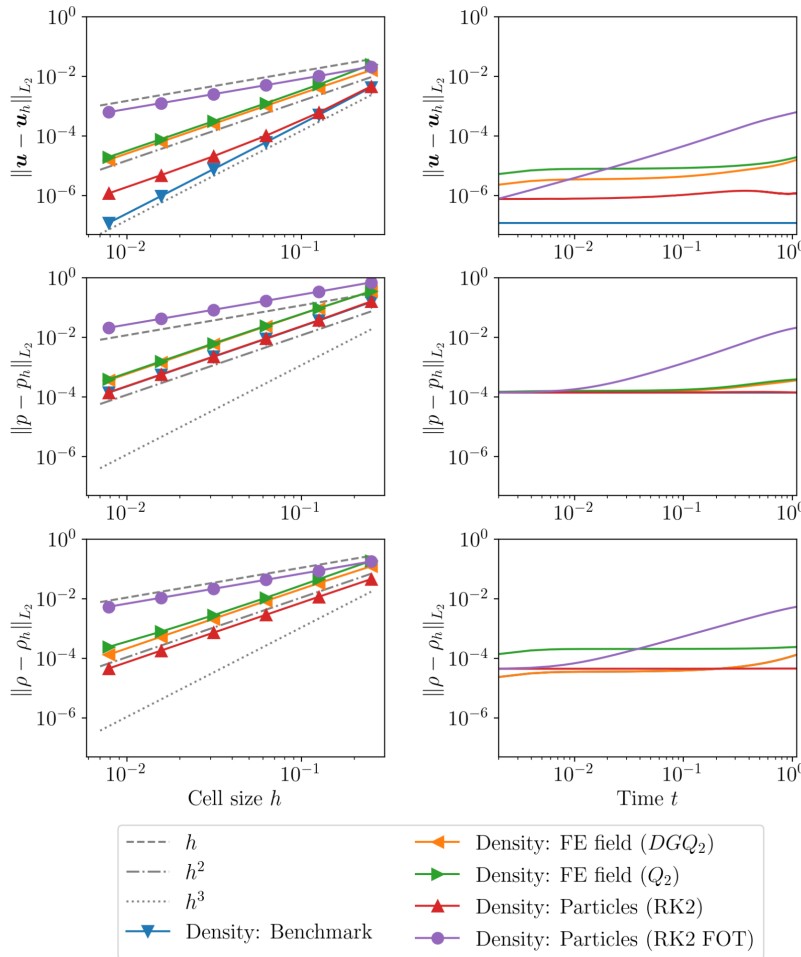

**Figure 4.** *Transient box benchmark. Left: $L_2$ error norms of velocity (top row), pressure (middle row), and density (bottom row) for different cell sizes h at time $t = \ln(1+2) \approx 1.0986$. Different colors and marker styles show different advection methods; gray lines show ideal first, second and third order convergence. Right: $L_2$ error norms of velocity (top row), pressure (middle row), and density (bottom row) as a function of time for resolution $h = 1/128$.*

When evaluating the error norms of the solution as a function of time for a fixed resolution (right column of Fig. 4), we can gain additional insight into the properties of the advection methods. While it is obvious that the RK2FOT methods remains the most inaccurate method at a sufficiently large time, this comparison also makes clear that it shares the same error value as RK2 at the start of the model, which is lower than the $Q_2$ and $DGQ_2$ methods. This is because the error in the first time step is dominated by the accuracy of the spatial approximation of the density. This also means that in benchmarks that are instantaneous or very short, the RK2FOT method will perform at nearly the same accuracy as the RK2 method, leading to misleading conclusions about its accuracy and suitability for time-dependent geodynamic models. Both $Q_2$ and $DGQ_2$ methods start at significantly larger errors in velocity and pressure, but accumulate less error over time than RK2FOT, although more than RK2.

For our conclusion it is important to note that even though both particle methods start with a significantly smaller error than finite-element advection methods, the first order accuracy of the RK2FOT scheme produces significantly larger errors and that this effect becomes more pronounced with increasing resolution and increasing model runtime.

## 5.3   Accuracy and performance discussion

Summarizing the benchmark results, first-order particle methods yield larger errors than the tested field-based methods, while higher-order particle methods outperform the investigated field-based methods both with increasing resolution and with increasing model time. Therefore, whenever other error sources of the solution are sufficiently small (i.e., if the Stokes element and time-stepping scheme allow for higher order accuracy) a higher order particle scheme can significantly enhance the accuracy of the solution. Even though we cannot prove it here, this conclusion is likely also true for the common case of a solution that is not smooth enough to allow for the optimal convergence rate of RK2. For discontinuous solutions the convergence rate of higher order finite elements can break down to the same rate as for first order elements (e.g. Thielmann et al., 2014). However, solutions are rarely fully discontinuous and instead contain a mix of smooth and non-smooth regions. Additionally, despite showing the same convergence rate, higher-order elements have still delivered higher accuracy in absolute terms than lower-order methods in many benchmarks (Kronbichler et al., 2012; Thieulot and Bangerth, 2022). We speculate that the same results would be seen for higher-order advection methods in time, although the construction of appropriate benchmarks would be challenging.

Finally, the improved accuracy of higher-order particle methods has to be discussed in the context of their larger memory requirement and computational cost. RK2 requires the storage of two velocity solutions instead of a single solution like RK2FOT; thus very coarsely (neglecting the memory cost of the particles) one could consider RK2 to be twice as expensive in terms of memory. However, this additional cost has to be compared to the total memory requirement of a modern geodynamic model and is only relevant if models are typically limited by the available memory. Many modern Stokes solvers in geodynamics either rely on matrix-based algorithms or Krylov subspace solvers with a long recurrence relation (e.g. GMRES), both of which can easily require the memory of tens to hundreds of solution vectors. For all of these models storing an additional velocity solution for the particle advection represents a negligible memory cost. Even if models are computed with modern matrix-free solvers with short-recurrence relations (e.g. Clevenger and Heister, 2021) in our experience the size of models are typically

**Table 1.** Performance comparison of RK2 and RK2FOT for the box benchmark with a shortened end time of $t = 0.1$ in dependence of resolution. The table shows the total model runtime (Model) and particle advection time (RK2 and RK2FOT) for the two advection algorithms, as well as the fraction of total model runtime used for particle advection (RK2 / Model and RK2FOT / Model) and the relative performance cost of RK2 vs RK2FOT. Note that parts of the algorithm like sorting particles into mesh cells or interpolating properties back to the mesh are not listed, because their cost is independent of the advection algorithm.

| Cells | #Steps | #Particles | Model | RK2 | RK2 / Model | Model | RK2FOT | RK2FOT / Model | RK2 / RK2FOT |
| Units | - | - | [s] | [s] | [%] | [s] | [s] | [%] | [%] |
| --- | --- | --- | --- | --- | --- | --- | --- | --- | --- |
| $4^2$ | 3 | 2048 | 0.136 | 0.00158 | 1.1 | 0.133 | 0.00136 | 1.0 | 116.1 |
| $8^2$ | 5 | 8,192 | 0.248 | 0.00609 | 2.4 | 0.25 | 0.00552 | 2.2 | 110.3 |
| $16^2$ | 8 | 32,768 | 1.02 | 0.034 | 3.3 | 1.01 | 0.0279 | 2.7 | 121.8 |
| $32^2$ | 15 | 131,072 | 6.71 | 0.244 | 3.6 | 6.81 | 0.212 | 3.1 | 115.0 |
| $64^2$ | 28 | 524,288 | 48.5 | 1.81 | 3.7 | 48 | 1.59 | 3.3 | 113.8 |
| $128^2$ | 54 | 2,097,152 | 381 | 13.6 | 3.5 | 377 | 11.8 | 3.1 | 115.2 |
| $256^2$ | 107 | 8,388,608 | 3077 | 108 | 3.5 | 3056 | 94.1 | 3.1 | 114.8 |

All performance results were computed on one core of an AMD EPYC 7453 processor with the software listed in the Data Availability Statement.

limited by the available compute power or memory bandwidth, but rarely by the available total memory. We therefore assume that storing an additional velocity solution is not a prohibitive cost and focus for the rest of this section on investigating the performance of RK2 over RK2FOT.

In theory, and on the most granular level, RK2 could be expected to require 50% more memory bandwith than RK2FOT, because simplistically speaking its second stage computes $\mathbf{x}^{n+1} = \mathbf{x}^n + \frac{\Delta t}{2} \left( \mathbf{u}^n + \mathbf{u}^{n+1} \right)$ (reading three vector entries from memory) instead of $\mathbf{x}^{n+1} = \mathbf{x}^n + \Delta t \, \mathbf{u}^n$ (reading two). If the algorithm is bandwidth limited it could therefore incur a 33% performance penalty. However, this is of course only a small part of the total particle advection cost, which also includes algorithms that are independent of the integration scheme (like the first Runge-Kutta stage, or determining the interpolation functions from grid to particle location). Finally, it is important to consider what fraction of the total model cost is used to advect the particles, as even if RK2 is significantly more expensive than RK2FOT, it may deliver higher accuracy for a small fraction of the total model cost.

Table 1 illustrates these metrics for a version of the box benchmark that was shortened to a model run time of $t = 0.1$. As can be seen particle advection only requires an almost insignificant percentage of the total model runtime (always $\leq 3.7\%$). Additionally, the RK2 advection algorithm only incurs an additional cost compared to RK2FOT on the order of $0.4\%$ of the total model runtime or a relative increase of $10\%$ to $21\%$ compared to the cost of RK2FOT. Moreover, it is clearly visible that this additional cost is approximately constant across resolutions, which means that for a constant and relatively small additional cost these models deliver significantly more accurate solutions as shown in the previous sections. In our opinion, these results illustrate that as long as the memory cost of storing an additional velocity solution is acceptable, using RK2 with

an extrapolated or interpolated half-timestep is generally superior to a first order method like RK2FOT in applications that depend on accurate solutions.

## 6 Application: Evolution of the mineral grain size below oceanic lithosphere

Above we have illustrated the influence of algorithmic choices on the accuracy and performance of benchmark results. However, this does not by itself justify the increased cost of such an algorithm in practical models: perhaps, in typical geodynamic applications, the error due to a low-order time approximation is negligible compared to other error sources, and therefore a simple advection method may be sufficient. In the following, we use an application model to show that the higher accuracy is indeed important and can influence first-order outcomes and the interpretation of a geodynamic study.

In order to illustrate this point, we use an example where the property carried on the particles (the grain size $d$) nonlinearly influences the material properties (the viscosity $\eta$) and the corresponding solution of the equations. Our model setup (Fig. 5, top) consists of the oceanic lithosphere and asthenosphere, down to a depth of 410 km, moving away from a spreading center at the top left corner of the model and horizontally extending to a distance of 4000 km from the ridge. The grid does not change over time and is spatially uniform with a cell size of 12.8 km (for a total of 320 cells in $x$- and 32 cells in $y$-direction). The plate speed is prescribed in horizontal direction to 5 cm/yr at the top, and right (outflow) boundaries at depths smaller than 100 km. The outflow velocity then linearly decreases with depth starting at 100 km depth towards 0 cm/yr at the bottom of the model. The left (ridge axis) boundary of the model is closed and stress-free (free-slip). The vertical velocity component is not prescribed at the bottom and right boundaries, instead a depth-dependent hydrostatic pressure profile that is computed at the model start and is constant in time is prescribed. Therefore, material can flow in beneath the ridge axis and leaves the model either through the bottom or the right boundary. The initial temperature follows an adiabatic profile with a potential temperature of 1623 K, and a half-space cooling profile close to the top boundary with an age consistent with the plate velocity. The initial temperature also includes a small ($r = 10\,\mathrm{km}$) thermal perturbation at the ridge axis to support the onset of spreading. The initial grain size is set to $d = 5\,\mathrm{mm}$, and also includes a small ($r = 30\,\mathrm{km}$) Gaussian anomaly, which reduces the grain size to $d = 2.1\,\mathrm{mm}$ close to the ridge axis. Since the temperature is prescribed at the top boundary, the plate is cooling conductively over time until small-scale convection sets in at the base of the plate.

In this model, we use particles to carry information about the mineral grain size $d$, which influences the viscosity nonlinearly as

$$\eta_{\mathrm{diff}} = \frac{1}{2} A_{\mathrm{diff}}^{-1} d^m \exp\left(\frac{E_{\mathrm{diff}} + PV_{\mathrm{diff}}}{RT}\right), \tag{23}$$

$$\eta_{\mathrm{dis}} = \frac{1}{2} A_{\mathrm{dis}}^{-\frac{1}{n}} \dot{\varepsilon}_{\mathrm{dis,II}}^{\frac{1-n}{n}} \exp\left(\frac{E_{\mathrm{dis}} + PV_{\mathrm{dis}}}{nRT}\right), \tag{24}$$

resulting in an effective viscosity of

$$\eta_{\mathrm{eff}} = \frac{\eta_{\mathrm{diff}}\,\eta_{\mathrm{dis}}}{\eta_{\mathrm{diff}} + \eta_{\mathrm{dis}}}. \tag{25}$$

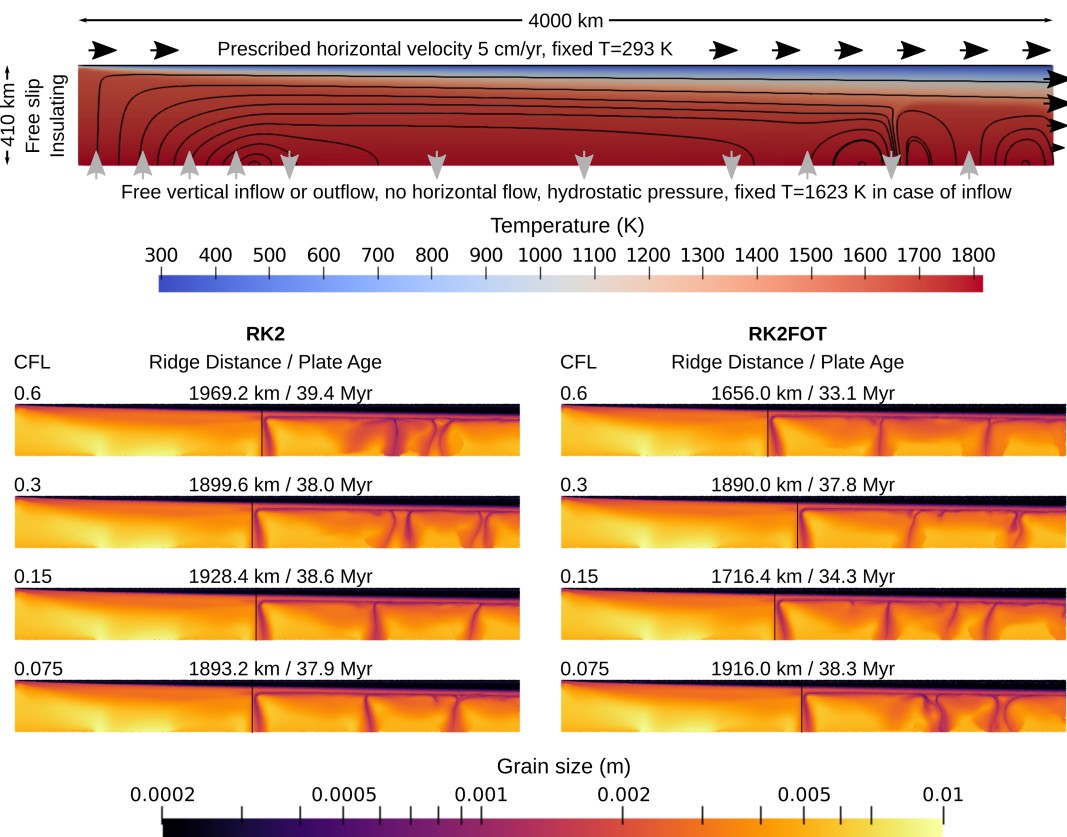

**Figure 5.** *Top: Setup of the application model. Background colors illustrate temperature, black solid lines are streamlines. Arrows indicate velocity. Bottom: Grain size at the end of the application model after 200 million years, for models using RK2 (left column), and RK2FOT (right column) for different time step lengths (rows). The onset of convection below the plate is marked for each model and measured as the distance from the ridge to the -50 K non-adiabatic temperature contour in 150 km depth. The corresponding age of the oceanic plate at this distance is also shown.*

Here, $\eta_{\text{diff}}$, $\eta_{\text{dis}}$, and $\eta_{\text{eff}}$ are the diffusion, dislocation and effective viscosity, respectively. $A_{\text{diff}} = 5 \times 10^{-15}$ m³/Pa/s and $A_{\text{dis}} = 10^{-15}$ Pa$^{-3.5}$/s are diffusion and dislocation creep prefactors, $E_{\text{diff}} = 375$ kJ/mol and $E_{\text{dis}} = 530$ kJ/mol the activation energy for diffusion and dislocation creep, $V_{\text{diff}} = 4 \times 10^{-6}$ m³/mol and $V_{\text{dis}} = 1.4 \times 10^{-5}$ m³/mol the respective activation volumes; $R$ the universal gas constant, $P$ the pressure, $T$ the temperature, $m = 3$ the grain size exponent of diffusion creep, $\varepsilon_{\text{dis,II}}$ the square-root of the second moment invariant of the dislocation strain rate, and $n = 3.5$ the dislocation creep strain rate exponent. We limit the viscosity computed in (25) to be between $10^{16}$ and $10^{23}$ Pa s.

In addition, particles are not just advected, but both the temperature and the strain rate in the model influence the evolution of the grain size. For a single particle moving along the flow field, we describe this evolution via the equation

$$\frac{d}{dt}d(t) = p_g^{-1} d^{1-p_g} k_g \exp\left(-\frac{E_g + PV_g}{RT}\right) - 4\dot{\varepsilon}_{\text{II}}\,\dot{\varepsilon}_{\text{dis,II}}\,\eta_{\text{eff}}\frac{\lambda d^2}{c\gamma}, \tag{26}$$

which implies that the grain size field $d(\mathbf{x}, t)$ satisfies the equation

$$\left( \frac{\partial d}{\partial t} + \mathbf{u} \cdot \nabla d \right) = p_g^{-1} d^{1-p_g} k_g \exp\left( -\frac{E_g + PV_g}{RT} \right) - 4\dot{\varepsilon}_{\mathrm{II}} \dot{\varepsilon}_{\mathrm{dis,II}} \eta_{\mathrm{eff}} \frac{\lambda d^2}{c\gamma}. \tag{27}$$

Here, $k_g = 1.92 \times 10^{-10}$ m$^3$/s is the grain size growth prefactor, $E_g = 400$ kJ/mol and $V_g = 0$ m$^3$/mol the activation energy and volume for grain size growth respectively. $p_g = 3$ is the grain size growth exponent, $c = 3$ a geometric constant, $\lambda = 0.1$ the fraction of deformation work that goes into changing the grain boundary area and $\gamma = 1$ the average specific grain boundary energy. The terms on the right-hand sides of these equations describe how the dynamic grain size increases over time (with a non-linear dependence on temperature and grain size itself) and how it is decreased by dynamic recrystallization due to strain accomodated by dislocation creep (which depends nonlinearly on stress and temperature). For a detailed discussion of these terms and all parameter values we refer to (Dannberg et al., 2017). We solve this model using the particle-based RK2 and RK2FOT advection schemes for different time step lengths expressed as fraction of the CFL number. We take care that except for the advection scheme all other parts of the model and algorithms are identical. In particular, we generate particles in identical and deterministic random locations, we enforce the same minimum (40) and maximum (250) number of particles per cell and make sure all algorithms for particle addition and deletion are deterministic.

As can be seen in the bottom panels of Fig. 5, the two advection methods produce noticeably different locations of onset of convection. While the model with a full RK2 advection scheme develops small-scale convection beneath the oceanic plate at a distance of approximately 1900 km from the ridge (from 1893.2 km to 1969.2 km), the model with RK2FOT develops the onset at varying distances from from 1656.0 km to 1916.0 km. These numbers correspond to plate ages of 37.9-39.4 Myr (RK2) and 34.3-38.3 Myr (RK2FOT), respectively. Because of the strong nonlinearity of these models we do not observe a simple convergence to one solution as in the benchmark results for either of the models. However, it is apparent that the RK2 method produces a much more stable onset location of small-scale convection, and a greater similarity of the other downwellings that develop beyond the initial onset. In contrast, the onset of convection varies significantly in the RK2FOT method depending on the time step size. In addition, the downwellings show very different convection structures. One could speculate that the RK2FOT method starts to converge towards the RK2 results for a CFL number of 0.075, however this is not certain given the strong variations in the RK2FOT results. Considering our benchmark results of the previous sections, we would expect the RK2FOT method to converge to the same solution as RK2, though requiring substantially smaller time steps; at least in this application it would clearly require time steps that make the model prohibitively expensive.

More importantly, one could assume that the shown variations just illustrate temporal variability in the convection pattern, and that the RK2FOT results are only a temporary state at the end of the model (200 Myr). To investigate this question we track the onset of convection over time for both RK2 and RK2FOT and show the results in Table 2. We find that the different outcomes between the advection methods is not just a temporary state at this precise time, but the models show systematic differences over long ranges of time. The onset of convection is relatively stable in the model with RK2, while it varies significantly in the model with RK2FOT and occurs systematically closer to the ridge as time passes. Even thought this may be a temporary development that will eventually reverse, we have not observed a similar behavior for RK2

**Table 2.** Onset of convection for RK2 and RK2FOT for the application model with a CFL number of 0.15 over time. The table shows distance from ridge and plate age of onset of convection in dependence of model time for both RK2 and RK2FOT.

| Method | RK2 | | RK2FOT | |
|---|---|---|---|---|
| Time | Distance | Plate age | Distance | Plate age |
| Myr | km | Myr | km | Myr |
| 120 | 1931.2 | 38.6 | 1888.4 | 37.8 |
| 140 | 1941.6 | 38.8 | 1855.6 | 37.1 |
| 160 | 1938.4 | 38.8 | 1843.2 | 36.9 |
| 180 | 1942.0 | 38.8 | 1827.6 | 36.6 |
| 200 | 1928.4 | 38.6 | 1716.4 | 34.3 |

The exact timing of onset of convection beneath an oceanic plate is relevant for the argument that small-scale convection causes a flattening of topography in seafloor subsidence datasets, and therefore ultimately for supporting the plate model of oceanic lithosphere cooling (Stein and Stein, 1992; Huang and Zhong, 2005; Richards et al., 2018). In addition to the difference in onset of convection, the characteristic length scale at which instabilities develop below the lithosphere is significantly smaller for the RK2FOT method, visible in the larger number and smaller distance between convection cells in the bottom panel
of Fig. 5. This is especially relevant as the distance between seismic anomalies associated with small-scale convection is a constraint from seismic studies and can be used to validate geodynamic models (Eilon et al., 2022). We want to emphasize here that our model is conceptual and not intended to produce realistic timings or length scales, but rather that a misprediction of these quantities due to inaccurate particle algorithms in models has concrete consequences for the interpretation of geodynamic features observed on Earth.

Because both the onset of small-scale convection and the length scale of convection cells is governed by the growth of small instabilities in a boundary layer, it is reasonable to assume that the lower accuracy of RK2FOT supports this growth of instabilities and explains the earlier onset of convection. The growth of instabilities in a boundary layer (or internally layered systems) is one of the most common processes for developing flow features in convecting systems like the Earth's mantle and lithosphere. Examples are the generation of plumes at the core-mantle boundary, the stagnation of subducted slabs or plumes at
phase transitions or the initiation of plate boundaries in models of lithosphere dynamics. We therefore infer from our results that models of all of these processes can benefit from incorporating more accurate particle advection methods, and that predictions of models using lower order advection schemes may need to be adjusted or reproduced in higher resolution studies.

## 7    Conclusions

We have shown in our benchmarks and applications that implementing accurate particle algorithms, in particular higher-order
in time, can significantly improve the numerical accuracy of geodynamic models. One of the conclusions of our benchmarks is

that commonly used particle advection methods that are higher order in space but first order in time acquire significant amounts of numerical error in time-variable flow, which becomes more pronounced the higher the resolution and the longer the model runs. The reason this error is not often discussed in the geodynamic literature is that traditional benchmarks that either rely on instantaneous analytical solutions, or on steady-state solutions, cannot show this error by their design. Only model comparison studies or benchmarks with analytical solutions in transient flow can point out this error source. Given that many geodynamic finite-element models already use Stokes elements that allow for higher order accuracy to ensure stability (e.g. Taylor-Hood $Q_2 \times Q_1$ or $Q_2 \times P_{-1}$), it would be straightforward to extend their particle advection algorithms to a second-order in time method. While this can increase the cost for evaluating velocities at the particle locations, our results show that the increased convergence order and improved accuracy of the model results is well worth the additional cost. Of course in order to increase the overall model accuracy, all other employed algorithms need to support the same accuracy.

We believe that a sharper focus on quantifying the numerical accuracy of geodynamic models will generate more trust in geodynamic model solutions and increase the impact of the discipline of geodynamic modeling as a whole. We provide the reference implementation of our algorithms and benchmarks in the open-source community software ASPECT and hope that they are useful to the community at large.

*Code availability.* Computations were done using the ASPECT code (Heister et al., 2017; Kronbichler et al., 2012; Bangerth et al., 2023, 2022). ASPECT is published under the GPL2 license, and the necessary software version and model configuration to reproduce the results is published on Zenodo as https://zenodo.org/records/10805269 (Gassmöller, 2024).

*Author contributions.* R.G. devised the study, devised and ran the benchmark cases, and wrote the majority of the manuscript. J.D. provided and described the grain-size application model. W.B. developed the integrator error analysis. E.G.P. provided the particle interpolation algorithm. E.G.P. and C.T. developed the instantaneous solution of the annulus benchmark case. All authors jointly interpreted the results and improved the manuscript.

*Competing interests.* The authors declare that they have no conflict of interest.

*Acknowledgements.* The authors acknowledge University of Florida Research Computing (http://researchcomputing.ufl.edu) for providing computational resources and support that have contributed to the research results reported in this publication. Computations were also run on the Stampede2 system at the Texas Advanced Computing Center (TACC) as part of award TG-EAR080022N.

We thank the Computational Infrastructure for Geodynamics (https://geodynamics.org) – funded by the National Science Foundation under awards EAR-1550901, and EAR-2149126 – for supporting the development of ASPECT.

R. Gassmöller and W. Bangerth were partially supported by the National Science Foundation under award OCI-1148116 as part of the Software Infrastructure for Sustained Innovation (SI2) program; and by the Computational Infrastructure in Geodynamics initiative (CIG), through the National Science Foundation under Awards No. EAR-0949446, EAR-1550901, EAR-2149126. W. Bangerth was also supported by the National Science Foundation under Awards OAC-1835673 and EAR-1925595.

E. G. Puckett was supported by the National Science Foundation under Award ACI-1440811 as part of the SI2 Scientific Software Elements (SSE) program.

J. Dannberg and R. Gassmöller were supported by the National Science Foundation under awards EAR-1925677 and EAR-2054605.

## Appendix A: Derivation of an exact solution in an annulus

Extending our previously published spherical benchmark (Gassmöller et al., 2019) seems to be straightforward by adding a time-dependent solid body rotation to the existing solution. However, because our earlier solution is already a purely rotational flow, an additional time-dependent rotation does not create a transient solution for the density, not allowing to intuitively measure the accuracy of the combined particle-finite-element algorithm. In other words, an error of the particle position along the streamline because of the time-variability of the flow would not change the density distribution and therefore would not translate into an error in the Stokes solution.

We therefore begin by deriving a new exact solution to the stationary, incompressible Stokes equations for an isoviscous, isothermal fluid in a two-dimensional annulus. Given the geometry of the problem, we work in polar coordinates. We denote the orthonormal basis vectors by $\mathbf{e}_r$ and $\mathbf{e}_\theta$, the inner radius of the annulus by $R_1$ and the outer radius by $R_2$. Further, we assume that the viscosity is a constant $\eta = 1$, and set the gravity vector to an inward-pointing vector $\mathbf{g} = -g_r \mathbf{e}_r$, with $g_r = 1$.

Given these assumptions, the incompressible Stokes equations in the annulus are (Schubert et al., 2001)

$$\frac{\partial^2 v_r}{\partial r^2} + \frac{1}{r}\frac{\partial v_r}{\partial r} + \frac{1}{r^2}\frac{\partial^2 v_r}{\partial \theta^2} - \frac{v_r}{r^2} - \frac{2}{r^2}\frac{\partial v_\theta}{\partial \theta} - \frac{\partial p}{\partial r} - \rho g_r = 0 \tag{A1}$$

$$\frac{\partial^2 v_\theta}{\partial r^2} + \frac{1}{r}\frac{\partial v_\theta}{\partial r} + \frac{1}{r^2}\frac{\partial^2 v_\theta}{\partial \theta^2} + \frac{2}{r^2}\frac{\partial v_r}{\partial \theta} - \frac{v_\theta}{r^2} - \frac{1}{r}\frac{\partial p}{\partial \theta} = 0 \tag{A2}$$

$$\frac{1}{r}\frac{\partial(rv_r)}{\partial r} + \frac{1}{r}\frac{\partial v_\theta}{\partial \theta} = 0. \tag{A3}$$

Equations (A1) and (A2) are the momentum equations in polar coordinates while equation (A3) is the incompressibility constraint.

We then seek solutions whose circumferential velocity can be written as

$$v_\theta(r,\theta) = f(r)\cos(k\theta) \tag{A4}$$

where $k$ is an integer and where $f(r)$ will be specified later. From equation (A3) we then obtain

$$\frac{\partial(rv_r)}{\partial r} = -\frac{\partial v_\theta}{\partial \theta} = kf(r)\sin(k\theta), \tag{A5}$$

leading to

$$v_r(r,\theta) = g(r)k\sin(k\theta), \tag{A6}$$

where

$$g(r) = \frac{1}{r} \int\limits^{r} f(r')dr'. \tag{A7}$$

Since we want to fix the velocity to be tangential at both boundaries we have

$$v_r(R_1,\theta) = v_r(R_2,\theta) = 0 \tag{A8}$$

for all $\theta \in [0,2\pi]$. We choose

$$f(r) = Ar + B/r \tag{A9}$$

in analogy to the solution of the Laplace equation in Chapter 6 of (Strauss, 2007), and thus

$$g(r) = \frac{A}{2}r + \frac{B}{r}\ln r + \frac{C}{r} \tag{A10}$$

where $C$ is a non-zero constant of integration. Given the boundary conditions in equation A8 we find that

$$A = -C\frac{2(\ln R_1 - \ln R_2)}{R_2^2 \ln R_1 - R_1^2 \ln R_2}, \qquad\qquad B = -C\frac{R_2^2 - R_1^2}{R_2^2 \ln R_1 - R_1^2 \ln R_2}. \tag{A11}$$

In this work we choose $C = -1$. Our earlier choice of $f$ means that

$$\frac{\partial^2 f}{\partial r^2} + \frac{1}{r}\frac{\partial f}{\partial r} - \frac{f}{r^2} = 0, \tag{A12}$$

so that equation (A2) simplifies to

$$\frac{1}{r^2}\frac{\partial^2 v_\theta}{\partial \theta^2} + \frac{2}{r^2}\frac{\partial v_r}{\partial \theta} - \frac{1}{r}\frac{\partial p}{\partial \theta} = 0, \tag{A13}$$

which together with equation (A4) leads to

$$p(r,\theta) = kh(r)\sin(k\theta) + l(r), \tag{A14}$$

where $l(r)$ comes from integration with respect to $\theta$ and $h(r) = (2g(r) - f(r))/r$. We now insert equation (A14) into equa-
tion (A1) to obtain

$$\begin{aligned}
\rho(r,\theta) &= \frac{\partial^2 v_r}{\partial r^2} + \frac{1}{r}\frac{\partial v_r}{\partial r} + \frac{1}{r^2}\frac{\partial^2 v_r}{\partial \theta^2} - \frac{v_r}{r^2} - \frac{2}{r^2}\frac{\partial v_\theta}{\partial \theta} - \frac{\partial p}{\partial r}\\
&= kg''(r)\sin(k\theta) + k\frac{g'(r)}{r}\sin(k\theta) - k^3\frac{g(r)}{r^2}\sin(k\theta)\\
&\quad - k\frac{g(r)}{r^2}\sin(k\theta) + k\frac{2f(r)}{r^2}\sin(k\theta) - kh'(r)\sin(k\theta) - l'(r),\\
&= m(r)k\sin(k\theta) - l'(r)
\end{aligned} \tag{A15}$$

 with

$$m(r) = g'' - \frac{g'}{r} - \frac{g}{r^2}(k^2 - 1) + \frac{f}{r^2} + \frac{f'}{r} \tag{A16}$$

$$= -\frac{f-g}{r^2} + \frac{f'-g'}{r} - \frac{f-g}{r^2} - \frac{g}{r^2}(k^2 - 1) + \frac{f}{r^2} + \frac{f'}{r} \tag{A17}$$

$$= -\frac{f-g}{r^2} + \frac{f'}{r} - \frac{f-g}{r^2} - \frac{f-g}{r^2} - \frac{g}{r^2}(k^2 - 1) + \frac{f}{r^2} + \frac{f'}{r} \tag{A18}$$

$$= -3\frac{f-g}{r^2} + \frac{f'}{r} - \frac{g}{r^2}(k^2 - 1) + \frac{f}{r^2} + \frac{f'}{r} \tag{A19}$$

since it is easy to verify using equation (A7) that $g'(r) = (f - g)/r$.

Taking $k = 0$ yields $\rho(r,\theta) = -l'(r)$, so we choose $l'(r) = -\rho_0$. In this case,

$$p(r,\theta)|_{k=0} = l(r) = \rho_0 \, g_r (R_2 - r), \tag{A20}$$

which represents a familiar linear pressure increase with depth for constant density and gravity, and where we have imposed $p(r,\theta) = 0$ at the outer radius $r = R_2$.

In summary, equations (A4), (A6), and (A14) form a solution of the incompressible Stokes equations, which fully stated reads

$$v_\theta(r,\theta) = f(r)\cos(k\theta), \tag{A21}$$

$$v_r(r,\theta) = g(r)k\sin(k\theta), \tag{A22}$$

$$p(r,\theta) = kh(r)\sin(k\theta) + \rho_0 g_r (R_2 - r), \tag{A23}$$

with

$$\rho(r,\theta) = m(r)k\sin(k\theta) + \rho_0, \tag{A24}$$

$$g_r = 1, \tag{A25}$$

$$g_\theta = 0, \tag{A26}$$

$$f(r) = Ar + B/r, \tag{A27}$$

$$g(r) = \frac{A}{2}r + \frac{B}{r}\ln r + \frac{C}{r}, \tag{A28}$$

$$h(r) = \frac{2g(r) - f(r)}{r}, \tag{A29}$$

$$m(r) = g''(r) - \frac{g'(r)}{r} - \frac{g(r)}{r^2}(k^2 - 1) + \frac{f(r)}{r^2} + \frac{f'(r)}{r}, \tag{A30}$$

$$A = -C\frac{2(\ln R_1 - \ln R_2)}{R_2^2 \ln R_1 - R_1^2 \ln R_2}, \tag{A31}$$

$$B = -C\frac{R_2^2 - R_1^2}{R_2^2 \ln R_1 - R_1^2 \ln R_2}, \tag{A32}$$

$$C = -1. \tag{A33}$$

We can use the velocity solution for $v_r$ and $v_\theta$ to determine a stream function for this flow field, which will be used to derive the stationary benchmark below:

$$\Psi(r,\theta) = -\left(\frac{A}{2}r^2 + B\ln(r) + C\right)\cos(k\theta). \tag{A34}$$

The solution above is time-independent and only valid for instantaneous models where the density is not advected. To make

it time-dependent, we first modify the density and gravity to create a steady-state variant of the benchmark and add a known time-dependent component to the velocity as described in Section 4. We start by choosing a density field consistent with the streamline $\rho(r,\theta) = \Psi(r,\theta)$. In our concrete benchmark this solution no longer satisfies the derived Stokes solution. However, we can recover an analytic solution by exploiting the fact that for the incompressible Stokes equations the density $\rho$ only enters the computation as a product with the gravity $\boldsymbol{g}$. Therefore, if as described in the example case above $m(r)k\sin(k\theta)$ is the

right hand side force term that satisfies the Stokes equation, we can still choose the density arbitrarily (e.g. $\rho(r,\theta) = \Psi(r,\theta)$), as long as we define the gravity to be $\boldsymbol{g}(r,\theta) = m(r)k\sin(k\theta)/\rho(r,\theta)$. This keeps the original forcing term constant, and so makes the solution independent of time. The steady-state solution therefore is the same as above, except:

$$\rho(r,\theta) = \Psi(r,\theta), \tag{A35}$$

$$g_r(r,\theta) = m(r)k\sin(k\theta)/\rho(r,\theta), \tag{A36}$$

with all other constants chosen as before.

In order to transform this steady-state benchmark into a known transient solution we then add a solid body rotation with a nonlinear time-dependent rotational velocity to the flow field. Since solid body rotations lie in the nullspace of the incompressible Stokes equations on an annular domain, the resulting flow field will still be a solution of the incompressible Stokes equations. This approach will work as long as we perform an appropriate rotation of all components of the solution, and is

equivalent to defining the solution in a rotating reference frame. We therefore modify the velocity components in $\theta$ direction to

$$v_\theta(r,\theta,t) = f(r)\cos(k(\theta - \tau(t))) + r\omega(t). \tag{A37}$$

Here $\tau(t)$ is a phase shift, and $\omega(t)$ is an angular velocity. The phase shift $\tau(t)$ can be computed as the time integral of the angular velocity from the beginning of the model up to the present time $t$:

$$\tau(t) = \int_0^t \omega(s)\,ds. \tag{A38}$$

In order to not make the problem too simple, we forgo the case of a constant angular velocity and instead choose $\omega(t) = e^t$, resulting in $\tau(t) = e^t - 1$.

Since the modification of the velocity in equation (A37) by the solid body rotation $r\omega(t)$ lies in the nullspace of the Stokes equations, it is straightforward to compute the modifications of the remaining solution variables, which only involves adding

the phase shift to the $\theta$ coordinate.

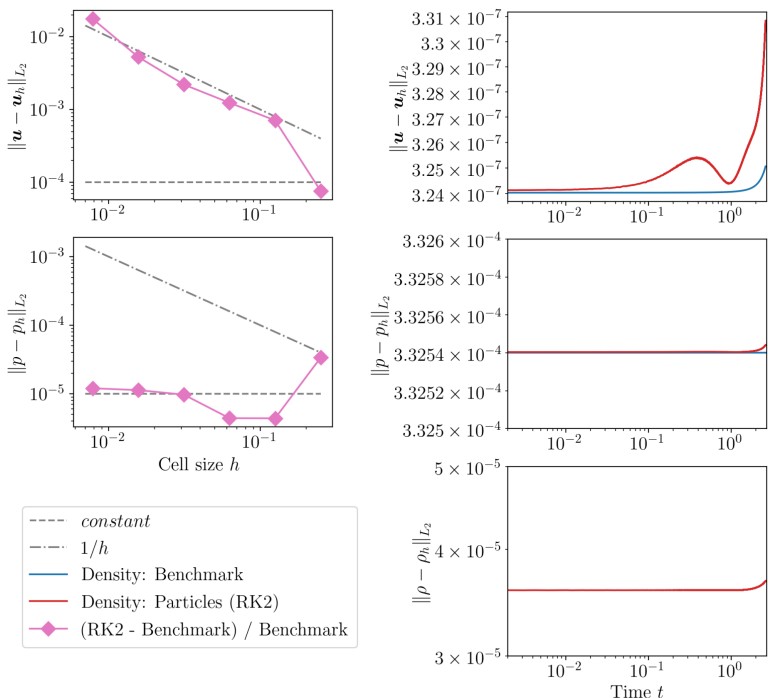

**Figure B1.** *Transient spherical annulus benchmark. Left: $L_2$ error norms of velocity (top row) and pressure (middle row) for different cell sizes h at time $t = \ln(1+4\pi) \approx 2.6075$. The pink line shows the relative difference in the error between RK2 and the analytical density model. Gray lines indicate the same convergence order (dashed line), or one convergence order lower (dash-dotted line) than the analytical density model. Right: $L_2$ error norms of velocity (top row), pressure (middle row), and density (bottom row) as a function of time for resolution $h = 1/128$ for analytical density and RK2 model.*

The final consideration is how to achieve this prescribed rotation in the model. Since in the incompressible Stokes equations stresses are transmitted instantaneously throughout the entire domain, we can use the exact, known velocities as boundary conditions and expect the motion to apply equally to the entire model domain.

## Appendix B: Detailed error investigation of the spherical shell benchmark

In order to better understand the accuracy of the RK2 method and investigate the source of the error decrease late in the model we show in Fig. B1 a detailed comparison of only RK2 against the analytical density method. In order to visualize the difference between RK2 and analytical density over resolution (left column) we no longer plot the absolute error in the $L_2$ norm, but instead the relative difference in error between RK2 and analytical density; i.e., if $\epsilon_{RK2} = \|\boldsymbol{u} - \boldsymbol{u}_h^{RK2}\|_{L_2}$ and $\epsilon_{AD} = \|\boldsymbol{u} - \boldsymbol{u}_h^{AD}\|_{L_2}$ then we plot $(\epsilon_{RK2} - \epsilon_{AD})/\epsilon_{AD}$. This way of plotting the error illustrates if both error values converge

at the same rate – leading to a constant relative difference between the two errors – or if the RK2 error indeed converges at a lower rate – leading to a linearly (or higher-order) increasing relative error difference towards smaller $h$. As it turns out the

relative error indeed increases linearly with resolution, meaning that RK2 converges at only second order, however the second order contribution is so small that it is not yet visible in the corresponding plot of Fig. 3. On the other hand, the pressure (left column, middle row), converges at the same second order rate for both analytical density and RK2, leading to a constant relative difference between the two errors.

Turning to the evolution of error over model time (Fig. B1, right column) reveals that what looked like a constant error value in Fig. 3 indeed follows the same trends as the other methods, only at drastically reduced error values. While the error values for velocity and density increased by 1-2 orders of magnitude from model start to end for the other advection methods (RK2FOT, $Q_2$, $DGQ_2$), they increase by at most $\approx 2\%$ for RK2. Additionally, RK2 features the same error reduction close to $t = 1$ as the other methods. Finally, it becomes apparent that even the model using analytical densities features a small but growing velocity error. Because the density error in this model is zero (analytical density) it seems reasonable to assume this error is a result of the Stokes solver. The accuracy of the Stokes solver depends on the absolute value of velocity, which increases exponentially over time. In other words, the blue line in the top right panel of Fig. B1 represents the best possible accuracy any advection method could reach for the given Stokes solver if it transported material information with perfect accuracy. Considering all the results presented in this section, we consider the RK2 scheme to be very close to achieving this theoretical limit.

To understand the reduction in velocity error and density error at certain model times requires us to take a closer look at the benchmark solution. Particularly relevant is that the benchmark solution is rotation-symmetric, with 4 regions of upwelling and 4 regions of downwelling. Therefore, rotating the density field by 90 degrees at any given time would lead to exactly the same solution. For some reason the reduction in velocity error in RK2 coincides almost exactly with a quarter rotation of the model solution at $t = \ln(1 + \pi/2) \approx 0.944$, the reduction for $Q_2$ coincides with a rotation by three quarters of a rotation ($t = \ln(1+3\pi/2) \approx 1.742$), and the reduction for RK2FOT coincides with a full rotation ($t = \ln(1+2\pi) \approx 1.985$). We speculate that the times of rotation symmetry with the starting solution allows for a resonance between the accumulated error in the numerical solution and the analytical solution at a slightly different time. This interaction would allow for an apparent reduction in error that does not actually exist, which is consistent with the observation that all errors rapidly increase again after the minimum. However, while this theory explains why reductions in error could happen at specific times, we have no explanation why the anomaly happens at different multiples of the rotation symmetry for different advection methods. We can only speculate that the occurence depends on a very specific feature in each model, for example how close individual discrete time steps end at the analytically determined times of rotation symmetry. Independent of the origin of the anomaly, the results of the convergence studies show that it does not influence the measurement of the convergence orders of different methods.

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
