# Peer review of "Benchmarking the accuracy of higher order particle methods in geodynamic models of transient flow"

_EGUsphere, 2023_

## Referee Comment (RC1)

**Review of Gassmöller et al. *Benchmarking the accuracy of higher order particle methods in geodynamic models of transient flow**

This is a well-written article that provides valuable insights into the source and time-evolution of the errors introduced in forward models by commonly used advection methods in geodynamic modeling software.

The study illustrates the higher accuracy of higher order particle methods for transient flow, and how this can potentially lead to significantly different results in geodynamic modeling applications compared to lower order methods. The higher order method proposed in this paper is also highly appealing as it requires minimal code modifications for those codes already using Particles-in-Cell methods to advect material properties.

This article also showcases the importance of benchmarking in numerical modeling and presents two new very interesting and well-thought benchmarks from which the existing software can benefit, as they can easily be implemented and reproduced in other existing codes.

For the reasons stated above, the article certainly deserves to be published in GMD. Please find below a set of minor general and in-line comments for the authors consideration.

**General comments**

**(1)** The integration method presented here as RK2 and RK2FOT is a particular case of the general two-stage Runge Kutta method
$$x \leftarrow x + \Delta t((1 - (1/2\alpha))\, u(t\,,x) + (1/2\alpha)\, u(t + \alpha\Delta t, x + \alpha\Delta t\, u(t\,,x)))$$
where $\alpha = 0.5$, known as the *midpoint method*. This should be noted somewhere when the different integration methods are described and discussed.

In my numerical codes I typically use $\alpha = 2/3$ (Ralston's method), which in my tests seemed to produce (slightly) better results and lower particle injection rates, compared to the midpoint method. Have the authors tested other values of $\alpha$ different from 0.5?

As the existing code would require minimal modifications to accommodate the general RK2 method, it would be worth it to test (and quantify) whether other RK2 variations produce more accurate results or not during the model time evolution, or whether we should avoid some values of $\alpha$. Additionally, the extra computational time should also be negligible when compared to the midpoint method.

**(2)** Many of the conclusions discussed in the last paragraphs of Subsections 5.1 and 5.2 are very similar. Perhaps it would be a good idea to merge them in a small Subsection 5.3, to avoid redundancy.

As later mentioned in one of my comments below, I believe it's important to quantify the performance of RK2 vs RK2FOT, instead of vaguely stating that RK2 is somewhat slower.

This will give a better idea of the trade-off between accuracy and speed, and perhaps help other code users in deciding which method to use based on the application of their models and their computational resources. Plots of performance vs time could be presented and discussed also in this new subsection. It is also stated that higher accuracy particle methods yield a faster solver convergence, if so, a comparison of the Stokes solver runtime for both cases can also be included here.

**(3)** I really like the two benchmarks in this paper. However, adding some more details about them in code may be beneficial. For example, the manuscript does not mention whether these benchmarks and models do inject or delete particles. If this is the case, it is worth briefly discussing whether RK2 and RK2FOT yield a similar or different number of particles at the end of the models, as well as the time-history of particle injection and deletion rates. The dimensions used in this article for the domain of these benchmarks should also be stated in the text.

**Line-by-line comments**

**L26:** It's clearly stated that the papers cited here are just some out of many. Nonetheless, some of the examples have a large collection of previous works that I think it would be appropriate to at least include a second reference. And also add (e.g. …) to the citation to emphasize the fact that they are just some of the many examples.

**L34-35**: Give some examples/references for every stated method. For example, ASPECT and Moresi et al. 2014 for FEM, Gerya and Yuen 2003 and Kaus et al. 2016 for FDM, or Tackley 2008 for FVM, among others.

**L39**: I think it is better to mention it is "application agnostic" a bit below, where it is described that the particle infrastructure is implemented in a general purpose FEM library such as deal.II, rather than in ASPECT itself. Is this particle architecture specific to FEM models? For example, can it deal with a finite difference code with a staggered grid?

**L39**: most modern HPC centers offer a considerable amount of GPU resources. Does the software here presented run on a (multi)GPU environment? If not, please clarify in the text.

**L54**: *"...due to interpolating properties from particles to **fields**"*
I think using "grid" or "mesh" may be more clear than "fields"

**L80**: Since you write the vector fields in bold, mention this fact in the lines below.

**L82**: strain rate tensor

**L88**: $N$ is not explicitly defined.

**L108**:  values

**L109**:  result

**L109**:  (2)

**L111**: Why use *j* as a sub-index? *i* is not used elsewhere for any other purpose as a sub-index, and it seems a more obvious choice since the particle information is likely to be stored in 1D arrays.

**L119**: *"the equation the differential equation solver really tries to solve is"*
I find this phrasing a bit weird, I suggest something in the line of "the equation being solved is actually", or similar.

**L130** I find somewhat random the use of inverted commas and italic font for the word exact.

**L145** *"although many other methods are available and have been used in geodynamic applications."*
Provide some examples/references.

**Eq 7.** Later on in equation (29), the total time derivative d/dt is not in italic. For consistency, denote d/dt everywhere either in italic or normal font.

**L223**: *"properties: First"*
"properties: first" or "properties. First"

**Section 4.1:** Specify, either in the main text or in the caption of Figure 1, what are the inner and outer radii of the spherical shell.

**L235**:  In other words,

**L248:**  Appendix A

**L266**: Shouldn't it be $x \leftarrow$ for the coordinate transformation?

**Equations 20, 21, 22, 23, 25** : all of these equations are missing a \right) bracket to close the first trigonometric function. E.g, eq. (20) should be $\sin(\pi(x - \tau(t)))$ instead of $\sin(\pi(x - \tau(t))$

**L287**: Backward Differentiation Formula (BDF2)

**L295**: Perhaps add a brief comment on the Q2 x Q1 element, as you do for Q2 and DGQ2. I believe Q1 is not mentioned elsewhere.

**L296**: Clarify here whether there is injection and removal of particles. And if this is the case, please mention what are the minimum and maximum number of particles per element.

**L311:**  The analysis

**L314**: What is the real difference in % of the error of RK2 between the first and last time step?

**L314**: *"While RK2 and RK2FOT start off at the same error value, and RK2 almost maintains this error over the evolution of the model, the error of RK2FOT increases significantly over time."*

May benefit from some rephrasing. Along the lines of *"Both RK2 and RK2FOT start with the same error value and, while RK2 error remains near-constant over the evolution of the model, the error of RK2FOT increases significantly with time."*

**L329-330**: The extra computation that RK2 is doing $u(t + dt/2) = (u^n + u^{n+1}) * 0.5$ which is $(2\ mem\ read + 1\ add + 1\ multiply) * ndim$. Where the slow part is the extra mem reads, and the extra flops are likely to be virtually for free. Thus, the price of doing $u^n * dt * 0.5$ in RK2FOT indeed doubles, give or take. However, unless I am missing something or there is something else going on, this still appears to be far from doing 2x operations during the entire integration algorithm, compared to RK2FOT. Could you please elaborate on this point?

Additionally, it would be beneficial to quantify the overhead of RK2 in comparison to RK2FOT. A Plot could be added, to compare and discuss the time evolution of the performance of both methods (shown as [runtime RK2] / [runtime  RK2FOT]) for different grid sizes. This will also show whether the performance of RK2 deteriorates with time more or less than in RK2FOT.

Last, doesn't RK2 have a higher memory footprint since it needs the storage of two velocity solutions?

**Section 6:** Some information is missing in the description of model setup. What is the resolution of this model? is the mesh regular or spatially refined? is AMR active? I found this information in the .prm files, but it would be useful to add it in the manuscript as well, since not every reader will check the input files.

**L369**:  perhaps

**L370:**  sources, and therefore

**L377**: *"…, and right (outflow) boundary and linearly decreases with depth starting at 100 km towards 0 cm/yr at the bottom of the model;..."*
I find this sentence a bit confusing. I guess the authors mean that there is a prescribed outward velocity at the right hand side boundary, which is constant for depth < 100km, and then decreases linearly to 0 at the bottom. However, the arrows in the right boundary of the model in the top panel of Figure 5 appear to be of the same length, top to bottom. It is worth rephrasing it in a clearer manner.

**L369:**  boundaries

**L383:** what is the magnitude of the anomalies? is it a Gaussian shape or constant value

**L402**: is "field" really meant to be in italic?

**L407:** *"For a detailed discussion of these terms and all parameter values we refer to (Dannberg et al., 2017)."*
It may make sense to move this sentence at the end of the very same paragraph.

**L441:** model  runs

**Figure 2:**
- Looks like the aspect ratio of the model is 1:2 but this is not written anywhere. You could just add the corner x and y coordinates in one (or all) of the plots. Or just write it down in the caption
- The ticks of both colorbars in the upper plots display some integers as proper integers (e.g. "**2"**) and others as floats (e.g. "**1.**"). For consistency, I suggest displaying all the integers as proper integers, as in Figure 1.

**Figure 5:**
- Would be nice add a video of these two models to the Zenodo repository
- Better to write "Plate age" in the titles of the middle and bottom panels, instead of just "Age"
- Light gray text and arrows are not the easiest to read on top of a white background. I would suggest the use of a different (darker) color

**Code availability:**
- I haven't tried to install ASPECT and run these benchmarks and models myself, but it looks like all the necessary files to reproduce the results in sections 5.1, 5.2 and 6 are indeed included in the repository, along with a good and much appreciated description of where to find these files and (potential installation problems aside) how to run them.
- Perhaps add the link to the Zenodo repo to the manuscript to make it easy to the readers => https://zenodo.org/records/10161412
- I may be looking in the wrong place, but I don't see a 2.6.0-pre branch. And the given git hash seems to link to a unrelated commit (https://github.com/geodynamics/aspect/commit/299a6456385b1fde6564fc079f3aa01cac075f24)

**Other minor suggestions**

Here is a list of some minor style and writing suggestions that may improve the readability of some sentences. It is up to the authors of the manuscript to decide which suggestions they believe will improve the quality of the text:

**L21**: scalable, and efficient
**L22**: investigate  complex
**L42**: general purpose open source finite element  library

**L42**: to model a wide range of geoscientific applications
**L61**: "and therefore fast changes over time"
Is this missing a verb?
**L74**:
**L89**: , for example,
**L124** The particle positions contain error contributions from the…
**L138:**
**L146**:
**L149**: , in practice
**L150**: number of around
**L154**:
**L199**: (1), (2), and (4)
**L230**: geometry, and
**L234**: … for the density, not allowing to…
**L235**: in other words,
**L280**: in equations (15) and (23)
**L299**: In the following subsections,
**L312**: , as expected
**Fig.3 L3**: for  the exact
**L371**: …method may be sufficient

**References**

Moresi, Louis, Zhong, Shijie, Han, Lijie, Conrad, Clint, Tan, Eh, Gurnis, Michael, Choi, Eunseo, Thoutireddy, Pururav, Manea, Vlad, McNamara, Allen, Becker, Thorsten, Leng, Wei, & Armendariz, Luis. (2014). CitcomS v3.3.1 (v3.3.1). Zenodo. https://doi.org/10.5281/zenodo.7271920

Gerya, T.V. and Yuen, D.A., 2003. Characteristics-based marker-in-cell method with conservative finite-differences schemes for modeling geological flows with strongly variable transport properties. *Physics of the Earth and Planetary Interiors*, *140*(4), pp.293-318.

Kaus, B.J., Popov, A.A., Baumann, T., Pusok, A., Bauville, A., Fernandez, N. and Collignon, M., 2016, February. Forward and inverse modelling of lithospheric deformation on geological timescales. In *Proceedings of nic symposium* (Vol. 48, pp. 978-983). John von Neumann Institute for Computing (NIC), NIC Series.

Tackley, P.J., 2008. Modelling compressible mantle convection with large viscosity contrasts in a three-dimensional spherical shell using the yin-yang grid. *Physics of the Earth and Planetary Interiors*, *171*(1-4), pp.7-18.

---

## Author Comment (AC1)

**Reply to reviewers**

Below, please find a reply to the points raised by the reviewers. Reviewer comments are in blue, our replies in black. Issues brought up in reviews but not specifically discussed below have simply been fixed in the revision as suggested by the reviewers.

We would like to express our gratitude to the reviewers for the very careful reading of our manuscript, along with the thoughtful comments. Peer review makes papers better, and this observation applies to the current one as well.

**Reviewer #1**

Review of Gassmöller et al. Benchmarking the accuracy of higher order particle methods in geodynamic models of transient flow

This is a well-written article that provides valuable insights into the source and time-evolution of the errors introduced in forward models by commonly used advection methods in geodynamic modeling software.
The study illustrates the higher accuracy of higher order particle methods for transient flow, and how this can potentially lead to significantly different results in geodynamic modeling applications compared to lower order methods. The higher order method proposed in this paper is also highly appealing as it requires minimal code modifications for those codes already using Particles-in-Cell methods to advect material properties. This article also showcases the importance of benchmarking in numerical modeling and presents two new very interesting and well-thought benchmarks from which the existing software can benefit, as they can easily be implemented and reproduced in other existing codes.
For the reasons stated above, the article certainly deserves to be published in GMD. Please find below a set of minor general and in-line comments for the authors consideration.

General comments

(1) The integration method presented here as RK2 and RK2FOT is a particular case of the general two-stage Runge Kutta method
$x \leftarrow x + \Delta t( (1 - (1/2\alpha)) u(t, x) + (1/2\alpha) u(t + \alpha\Delta t, x + \alpha\Delta t u(t, x)))$
where α = 0.5, known as the *midpoint method*. This should be noted somewhere when the different integration methods are described and discussed.

Yes, good suggestion. We added the name where we describe the method, in Section 3.3.

In my numerical codes I typically use α = 2/3 (Ralston's method), which in my tests seemed to produce (slightly) better results and lower particle injection rates, compared to the midpoint method. Have the authors tested other values of α different from 0.5?

As the existing code would require minimal modifications to accommodate the general RK2 method, it would be worth it to test (and quantify) whether other RK2 variations produce more accurate results or not during the model time evolution, or whether we should avoid some values of α. Additionally, the extra computational time should also be negligible when compared to the midpoint method.

We appreciate the comment and have done some tests on the generalized RK2 method as suggested. We indeed find for RK2FOT that Ralston's method decreases the error in our benchmarks slightly (by about 1% of the error, independent of resolution). However, we encountered two difficulties that have convinced us to limit ourselves to the midpoint method for our practical implementation:

- For RK2, the benefit of Ralston's method in our implementation depends on the resolution and systematically changes from about 1% (as in RK2FOT) at coarse resolutions to -1% at fine resolutions. This is surprising and we have carefully made sure it is not a mistake in our implementation. We assume that this error increase stems from the fact that we do not recompute the Stokes solution at the intermediate time, but instead extrapolate from the past two timesteps (as described in line 193-194 of the manuscript). Therefore the benefit of the reduced truncation error is counteracted by the disadvantage of increasing the extrapolation error from t+0.5dt to t+(2/3)dt. Of course recomputing the Stokes equation at the intermediate time would resolve this problem, but the cost would be significantly larger than the overall particle advection algorithm.
- While the extra computational time of the generalized RK2 is indeed negligible (or zero), the method requires more memory per particle than the midpoint method. In the midpoint method the increment $k1$ is conveniently no longer required after the first stage of the RK2 method. In contrast, in Ralston's method, both $k1$ and $k2$ are needed to compute the final updated position. This is important for us, because our particle advection happens stage-by-stage instead of particle-by-particle. That is, our particle advection algorithm works as follows:
  - All particles are advected through the first stage of RK2.
  - All particles that have left their old cells are sorted into their new cells.
  - (In a parallel computation) All particles that have left their MPI domains are transferred to neighboring MPI domains.
  - Then the advection of the second RK2 stage happens.
  - All particles that have left their old cells in the second stage are sorted into their new cells.
  - (In a parallel computation) All particles that have left their MPI domains in the second stage are transferred to neighboring MPI domains.

  Because of this algorithm all intermediate data that is required for the second stage has to be stored for all particles and shipped via MPI if necessary. For the midpoint method we only need to store x(t) and x(t+dt/2), while for the generalized method we need to store x(t), x(t+dt/2) and u(x,t), increasing the memory requirement by 50%.

None of these two reasons suggest the midpoint method is generally preferable, only that it is in practice perhaps somewhat easier to implement. We have added the following paragraph to the description of the RK2 method in section 3.3:

"Note that there are other RK2 methods, such as Ralston's method, which reduce the theoretical truncation error of the method while keeping the order of convergence. However, in our benchmarks the difference in error is small, and the midpoint method allows us to reduce the memory requirement of the algorithm."

(2) Many of the conclusions discussed in the last paragraphs of Subsections 5.1 and 5.2 are very similar. Perhaps it would be a good idea to merge them in a small Subsection 5.3, to avoid redundancy.

We agree and have merged the mentioned paragraphs in the new Subsection 5.3.

As later mentioned in one of my comments below, I believe it's important to quantify the performance of RK2 vs RK2FOT, instead of vaguely stating that RK2 is somewhat slower. This will give a better idea of the trade-off between accuracy and speed, and perhaps help other code users in deciding which method to use based on the application of their models and their computational resources. Plots of performance vs time could be presented and discussed also in this new subsection.

We followed this suggestion and quantified the performance cost in a shortened version of our box benchmark (only run to t=0.1). We discuss the results in the new Subsection 5.3, but to summarize here: In our implementation RK2 is approximately 20% slower than RK2FOT (when only comparing the advection cost), or about 0.4% slower in total model runtime. This performance cost stays constant across different resolutions of the model. Since the cost is constant we have refrained from adding a plot, and instead included a table of the model runtime and relative cost for both methods (Table 1).

It is also stated that higher accuracy particle methods yield a faster solver convergence, if so, a comparison of the Stokes solver runtime for both cases can also be included here.

We probably described that in a misleading way. What we meant was that the method shows a higher convergence order, therefore the error converges faster towards zero when increasing the resolution. We have removed this part of the sentence in question, since we mention the higher convergence order anyway.

(3) I really like the two benchmarks in this paper. However, adding some more details about them in code may be beneficial. For example, the manuscript does not mention whether these benchmarks and models do inject or delete particles. If this is the case, it is worth briefly discussing whether RK2 and RK2FOT yield a similar or different number of particles at the end of the models, as well as the time-history of particle injection and deletion rates. The dimensions used in this article for the domain of these benchmarks should also be stated in the text.

This was indeed an important oversight. We have added the relevant information in the location requested in the line-by-line comments. As explained in the new text, while our models enforce a minimum number of particles per cell by generating new particles if necessary, this never happens for the presented benchmarks. Based on our earlier work (Gassmöller et al, 2018) we would expect RK2FOT to require slightly more particle injections/deletions, however our earlier comparison was between forward Euler integration and RK2 and so we speculate that the

difference between RK2FOT and RK2 is significantly smaller for most models. The new paragraph reads:

"Because the number of particles in a cell can change during the model run, we enforce a minimum number of particles per cell of 12, which guarantees that the linear least-squares interpolation algorithm is always sufficiently constrained. We do not limit the maximum number of particles per cell in these models. In practice, the presented benchmarks never require the addition of particles, and therefore the number of particles stays constant (for the box) or decreases by less than 0.01% (for the annulus, caused by integration error close to the boundary, and then leading to the loss of particles across the boundary). The two tested integration schemes do not show a significant difference in particle loss in the annulus geometry, even though we have observed such differences between Runge-Kutta algorithms and Forward Euler integrators in our earlier work (Gassmöller et al., 2018)."

**Line-by-line comments**

L26: It's clearly stated that the papers cited here are just some out of many. Nonetheless, some of the examples have a large collection of previous works that I think it would be appropriate to at least include a second reference. And also add (e.g. …) to the citation to emphasize the fact that they are just some of the many examples.
Done.

L34-35: Give some examples/references for every stated method. For example, ASPECT and Moresi et al. 2014 for FEM, Gerya and Yuen 2003 and Kaus et al. 2016 for FDM, or Tackley 2008 for FVM, among others.
Done.

L39: I think it is better to mention it is "application agnostic" a bit below, where it is described that the particle infrastructure is implemented in a general purpose FEM library such as deal.II, rather than in ASPECT itself. Is this particle architecture specific to FEM models? For example, can it deal with a finite difference code with a staggered grid?
Thank you for the suggestion, we have moved the statement as suggested and specified that the algorithms itself are independent of the Stokes solution discretization. However, the algorithms are integrated into deal.II and exploit the mesh structure and mesh functions provided by deal.II, so transferring it to another code would still require replacing these parts.

L39: most modern HPC centers offer a considerable amount of GPU resources. Does the software here presented run on a (multi)GPU environment? If not, please clarify in the text.
We have specified that currently only CPU cores are supported. Since GPU support in both deal.II and ASPECT is in the early planning stage we do not consider this a major flaw in the particle algorithm at this point in time.

The new sentences reads: "In particular, it supports advanced computational methods such as an adaptively refined, unstructured, and dynamically changing background mesh, parallelization

beyond tens of thousand of (CPU) compute cores, storing arbitrary particle properties, and complex nonlinear solvers."

We agree that the current formulation is not ideal, but think "grid/mesh" is also too specific. We have changed the phrasing to "from particles to Stokes discretizations".

Good point, we used $i$ as a sub-index for another variable in an earlier version of the manuscript, but now it makes more sense to use $i$ for the particles. We have changed this in all relevant places.

We have changed the relevant sentence to:

"… although other methods are available and have been used in geodynamic applications (e.g. Heun's method in Zhong and Hager (2003); Sime et al. (2021), Runge–Kutta schemes with additional predictor-corrector steps Weinberg and Schmeling (1992), implicit Euler and BDF2 methods in Furuichi and May (2015), or Adams Bashforth methods in Adamuszek et al. (2016))."

"properties: first" or "properties. First"
Done.

Yes, that was an oversight. We have added dimensions to the setup figure as well as in the text for both the box and the spherical shell benchmark.

Thank you for the comment, this is indeed an important detail that we did not mention before. We have added the paragraph listed in the main comment above.

Added to the text: "Both RK2 and RK2FOT start with the same error value, but while the RK2 error remains near-constant over the evolution of the model (increasing by less than 2%), the error of RK2FOT increases by almost two orders of magnitude."

L314: "While RK2 and RK2FOT start off at the same error value, and RK2 almost maintains this error over the evolution of the model, the error of RK2FOT increases significantly over time." May benefit from some rephrasing. Along the lines of "Both RK2 and RK2FOT start with the same error value and, while RK2 error remains near-constant over the evolution of the model, the error of RK2FOT increases significantly with time."
We agree, changed.

L329-330: The extra computation that RK2 is doing $u(t + dt/2) = (un + un+1) * 0.5$ which is (2 *mem read* + 1 *add* + 1 *multiply*) * *ndim*. Where the slow part is the extra mem reads, and the extra flops are likely to be virtually for free. Thus, the price of doing $un * dt * 0.5$ in RK2FOT indeed doubles, give or take. However, unless I am missing something or there is something else going on, this still appears to be far from doing 2x operations during the entire integration algorithm, compared to RK2FOT. Could you please elaborate on this point?
Additionally, it would be beneficial to quantify the overhead of RK2 in comparison to RK2FOT. A plot could be added, to compare and discuss the time evolution of the performance of both methods (shown as [runtime RK2] / [runtime RK2FOT]) for different grid sizes. This will also show whether the performance of RK2 deteriorates with time more or less than in RK2FOT.
Last, doesn't RK2 have a higher memory footprint since it needs the storage of two velocity solutions?
Thank you for this comment; this sentence was indeed poorly formulated. What we meant was that in any individual stage of the algorithm we can imagine RK2 requiring at most 2x the operations or memory of RK2FOT (which is of course just an upper bound and not representative of actual total cost). The memory footprint between RK2 and RK2FOT is only identical for us because our implementation in ASPECT saves two Stokes solutions anyway, and the midpoint method requires the same amount of intermediate storage on particles for RK2 and RK2FOT. This is of course not true in general. We have clarified these points in the new subsection 5.3 with a more detailed discussion of total memory cost, required memory bandwidth, and actual measured performance. We also agree that a more quantitative and empirical performance comparison is useful. As discussed above it seemed more useful to us to include a table of the actual cost (relative to model runtime) and relative cost (relative to the other advection method) and have added the results in Table 1.

Section 6: Some information is missing in the description of model setup. What is the resolution of this model? is the mesh regular or spatially refined? is AMR active? I found this information in the .prm files, but it would be useful to add it in the manuscript as well, since not every reader will check the input files.
We agree, we added the following sentence in this section:

"The grid does not change over time and is spatially uniform with a cell size of 12.8 km (for a total of 320 cells in x- and 32 cells in y-direction)."

L377: "…, and right (outflow) boundary and linearly decreases with depth starting at 100 km towards 0 cm/yr at the bottom of the model;..."
I find this sentence a bit confusing. I guess the authors mean that there is a prescribed

outward velocity at the right hand side boundary, which is constant for depth < 100km, and then decreases linearly to 0 at the bottom. However, the arrows in the right boundary of the model in the top panel of Figure 5 appear to be of the same length, top to bottom. It is worth rephrasing it in a clearer manner.
We agree that this was not phrased very clearly, we have reformulated the sentence to:

"The plate speed is prescribed in horizontal direction to 5 cm/yr at the top, and right (outflow) boundaries at depths smaller than 100 km. The outflow velocity then linearly decreases with depth starting at 100 km depth towards 0 cm/yr at the bottom of the model."

L402: is "field" really meant to be in italic?
It was meant to emphasize the difference between Lagrangian and Eulerian perspective on the grain size, but it is not necessary, and we removed it.

Figure 2:
● Looks like the aspect ratio of the model is 1:2 but this is not written anywhere. You could just add the corner x and y coordinates in one (or all) of the plots. Or just write it down in the caption
We now added the model size both as labels in the figure and in the text.

Figure 5:
● Would be nice add a video of these two models to the Zenodo repository
We have added a video to the Zenodo repository.

● Light gray text and arrows are not the easiest to read on top of a white background. I would suggest the use of a different (darker) color
We have adjusted text and arrow color to increase the contrast.

Code availability:
● I haven't tried to install ASPECT and run these benchmarks and models myself, but it looks like all the necessary files to reproduce the results in sections 5.1, 5.2 and 6 are indeed included in the repository, along with a good and much appreciated description of where to find these files and (potential installation problems aside) how to run them.
Thank you

● I may be looking in the wrong place, but I don't see a 2.6.0-pre branch. And the given git hash seems to link to a unrelated commit (https://github.com/geodynamics/aspect/commit/299a6456385b1fde6564fc079f3aa01cac075f24 )
Except for the application model there is no special branch for this paper. All developed changes as part of this manuscript have been merged into ASPECT's main development branch, which was used to compute the models. We now removed the unclear sentence and instead refer to the Zenodo repository, which spells out which ASPECT version was used for which part of the study.

Other minor suggestions
[...]
We have incorporated most minor suggestions as suggested. Thank you.

**Reviewer #2**

General comments

I have read the manuscript of Gassmöller et al. with interest. The study investigates the accuracy of the particle in cell methods, widely used in geodynamic modeling. New analytical solutions are provided (and used) and I'm sure they will be very useful for the community. The main conclusion is that geodynamic simulators should include second-order in time particle advection solvers.

I think the manuscript is worth publication after addressing the moderate comments listed below. Addressing these comments should help with:
- providing a better overview of existing literature
- clarifying whether their conclusions are applicable in the general case (Main comment #1)
- demonstrating their points in a more convincing manner (Main comment #2)

**Section 1**

Reading the first part of the introduction gives the feeling that geodynamic modelling started in the lated 2010s. I would suggest to also cite older work. Numerical geodynamic models can be traced back to the late 1970s.
We intended to make readers aware of recent work, however you are of course correct that an introduction should give a broader overview. We have added references to earlier work.

l 36. you may want to oppose Eulerian to Lagrangian schemes (instead of field vs particle-based schemes).
We considered this suggestion, but would like to stick with the existing dichotomy of fields vs. particles. This is because there are both Eulerian and Lagrangian field-based methods (along with the "arbitrary Eulerian-Lagrangian (ALE)" perspective). In Lagrangian and ALE field-based methods, one moves the *nodes* of a finite element mesh, but the quantities of interest are still represented by fields, rather than located on points/nodes/particles.

l 37-46 the intro would better flow if this paragraph would be better located elsewhere (e.g. section 3). As such it reads weird. One would expect the intro to focus on the topic of advection (which is introduce in the line above this paragraph and discussed right after)
We agree that the current order of topics is not ideal. We have moved the paragraph in question into the Methods section, but left a sentence at the beginning of the last paragraph of the introduction that explains the motivation of our current study: "In this work, we measure the particle advection error in transient flow using the particle architecture we have developed as

part of our work on the Advanced Solver for Planetary Evolution, Convection, and Tectonics (ASPECT)."

l 58 yes, this is right, there is no systematic study of this type of error. Some examples are shown in Gerya and Yuen 2003 (e.g. Fig 12) and a potential solution is proposed (subgrid diffusion), the scheme is further extended to visco-elasticity in Gerya and Yuen (2007). This could be mentioned.
We agree that Gerya and Yuen 2003 show some good examples for this type of error and have included the reference in this sentence. However, we think there was a misunderstanding here, as we do not think that the subgrid diffusion algorithm is a potential solution for exactly the type of error we are thinking of in this sentence. Subgrid diffusion is a very interesting solution to the problem of stabilizing the properties (like temperature) that are carried on the particles by slowly diffusing them towards the interpolated solution (cell-wise linear or whatever other discretization is used). However, it cannot correct for any advection error in the particle position itself, which is what we are interested in here (it would be unclear what would be meant by diffusing the particle position towards a cell-wise interpolated solution).

I would mention Weinberg & Schmeling (1992) that uses particles with RK4 and a predictor-corrector scheme
https://www.sciencedirect.com/science/article/abs/pii/019181419290103-4

We agree, we have now mentioned Weinberg & Schmeling in the introduction about particle methods and in the list of different particle integrators in the methods section.

**Section 2**

eq 3 I'm not sure there is a point to include a diffusion term which is not used later on. Otherwise inactive terms could also appear in other governing equations (inertia, compressibility…).
Yes, this is a good point. We have simplified the description and omitted the intermediate step with the model that includes diffusion.

l 98 shear heating is not restricted to compressible flow. You probably want to refer to adiabatic heating instead.
Agreed, fixed.

**Section 3**

I would definitely put the paragraph about Aspect history here (it's currently in the middle of the introduction).
We agree this is a better place for the paragraph and have moved it. We have left a reference to our work on ASPECT in the introduction, as it provides the motivation for the current study.

In the description of each scheme (FE, RK2, RK2FOT) I would clearly indicate how many full (non-linear) mechanical solves are needed to integrate one time step.

We have now clarified that due to the choice of using already computed velocity solutions and if necessary interpolating between them, in our implementation the number of mechanical solves is actually independent of the chosen particle advection scheme. It only depends on the chosen nonlinear solver scheme for the Stokes equation, i.e. $u(t+\frac{1}{2}dt)$ is either extrapolated from previous timesteps (without nonlinear iterations), or interpolated between $u(t)$ and $u(t+dt)$ (with nonlinear iterations). Our previous phrasing did not make this clear enough. The sentence in the manuscript now reads:

"We will also assume that we have already solved the velocity field up to time $t(n+1)$ and are now updating particle locations from $x(n)$ to $x(n+1)$. In cases where one wants to solve for particle locations before updating the velocity field, $u_h$ can be extrapolated beyond $t(n)$ from previous time steps, or particle advection and velocity computation could be iterated in a nonlinear solver scheme. Because of this choice, the number of Stokes solutions which have to be computed is independent of the choice of particle advection scheme."

By reading the last paragraph, it may be worth mentioning somewhere the work of Furuichi and May (2015) on implicit advection:
https://www.sciencedirect.com/science/article/pii/S0010465515000569
There is also intersecting work on time integration of non-linear Stokes problems in Adamuszek et al (2016):
https://www.sciencedirect.com/science/article/abs/pii/S0191814116300013
Thank you for bringing these papers to our attention, they are indeed relevant and we have added them at an earlier point in the manuscript where we describe other integration schemes that have been used in the geodynamics community.

**Section 4.1**

I think both the analytic solution and its description are very useful.
Thank you.

**Section 4.2**

Here it reads a bit weird in the sense that the authors describe a modification of a solution used in a previous study and the difficulties encountered. As such it reads a bit like personal notes. Maybe you could state the complete solution in Cartesian coordinates directly and simply mention that it's different than the one used in the previous study? This would decrease the dependance of the current manuscript on this preceding study.
Upon reading Section 4.1 and 4.2 again we agree. We have shortened both sections by removing as many references to the earlier work as possible and stating the final solution earlier in each section. We have moved the description about the problems of using our earlier annulus benchmark into the appendix instead of removing it, as it provides the justification for why finding a new benchmark solution was necessary.

eq 19 is already defined in the text of section 4.1.
While shortening the sections we have removed this duplication.

**Section 5.1**

We agree that we should add a note here discussing this result. We have such a description in Section 5.2, where we can see an initial third-order convergence rate transfer into a second order rate for RK2 when the dominant error source changes from the spatial discretization to the particle advection. We would expect that the third order rate implies that RK2 advection is accurate enough so that other error sources dominate at the shown resolutions. We have added the following sentence to avoid repeating the discussion (which is better situated in the later section where we can actually observe the transition):
"The RK2 method maintains a third-order convergence rate in this metric up to very fine resolutions, which is surprising as the expected convergence order for RK2 is second order. We refer to Section 5.2 for a discussion on how RK2 may reach superconvergence for the resolutions shown in this benchmark."

We removed the sentence in question while addressing the comments of reviewer 1 and creating the new section 5.3 that includes a more detailed discussion on accuracy vs performance.

**MAIN COMMENT #1:**

We think that this is perhaps a good practical guideline, but in reality the situation is more complicated. As the formula at the end of Section 3.2 indicates, the total error is an interplay of spatial and temporal errors, plus the exponential growth of errors one incurs via Gronwall's Lemma. It would not be entirely unreasonable to suggest that for long-time simulations, a higher temporal than spatial order is appropriate in order to reduce (constant in) the exponential growth (e.g. RK4 could still deliver higher accuracy, even if it would not increase the convergence order compared to RK2).
In practice, the situation is made more complicated by the fact that typically not all advected variables are solved via particle schemes. The application in ASPECT, for example, typically has variables such as the temperature that also diffuse, and for these one will want to choose field-based methods that (for stability or accuracy purposes) require one to satisfy a CFL-type condition. For particle methods, it is perhaps not *necessary* to tie the time step to the spatial mesh size, but this is typically done nonetheless because it ensures, for example, that particles never move by more than one mesh size from their starting point, greatly simplifying the search for the new owner-cell in particle-in-cell schemes (and for all practical purposes making it possible to efficiently implement particle-in-cell schemes in parallel where one may also need to find the process owning a cell if it moves out of one partition). In other words, time step and

mesh size are typically chosen proportional to each other, and in that case it *does* make sense to choose at least equal temporal and spatial orders, in order to keep the computational cost balanced. Because we do not want to impose or prove an "optimal" combination of particle and Stokes solution methods we have left this discussion out of the main manuscript and instead only focus on the conclusion that it is worth considering higher-order particle advection methods.

My worry is that, in practice, model configurations generally include material boundaries with coefficient jumps (e.g. viscosity). In these cases, the order of accuracy of the mechanical solutions (L1, L2) will degrade (down to 1st order?) unless the jump is actually meshed. This last option is most often not selected in geodynamic models that rely on particles. Therefore, despite using second order stencils in space, the solutions are effectively of lower order. These cases are the most generally relevant but are not accounted for by the analytical solution nor in the next section.  So, in practice, is it still beneficial to use a second order time integration order if spatial accuracy is effectively of lower order?

Yes, the observation that the optimal spatial order is rarely achieved is probably true. It is definitely true for benchmark cases designed to mimic geodynamical situations, see for example Thieulot and Bangerth (2022):

https://www.math.colostate.edu/~bangerth/publications/2021-q1q1.pdf

In practice, however, we think that it is debatable whether the implication should be that one should just go with a lower-order scheme. As the mentioned paper shows, in situations where discontinuous coefficients reduce the convergence order, *higher-order methods are still substantially more accurate*. This is an observation we have also made in other contexts, and one can perhaps conjecture that the same is true in the time stepping context. It is worth mentioning that a spatial solution *u(x,t)* with reduced regularity (for which we obtain a lower spatial convergence rate) then appears as a coefficient in the particle trajectory; one would then expect also a lowered temporal convergence rate. Depending on the specific degree to which regularity is lost, the reduction in spatial and temporal convergence rates may or may not be the same.

It would make for an interesting study to test this systematically, but also one that will be quite difficult to design because it is likely very difficult to come up with analytic benchmarks. Absent concrete evidence to the contrary, our best (albeit educated) guess is that what works best for the cases with smooth solutions (as considered herein) is also going to work best for more realistic cases with reduced regularity.

We have integrated these thoughts into the following paragraph close to the beginning of the new Section 5.3:

"Summarizing the benchmark results, first-order particle methods yield larger errors than the tested field methods, while higher-order particle methods outperform the investigated field methods both with increasing resolution and with increasing model time. Therefore, whenever other error sources of the solution are sufficiently small (i.e., if the Stokes element and time-stepping scheme allow for higher order accuracy) a higher order particle scheme can significantly enhance the accuracy of the solution. Even though we cannot prove it here, this conclusion is likely also true for the common case of a solution that is not smooth enough to allow for the optimal convergence rate of RK2. For discontinuous solutions the convergence

rate of higher order finite elements can break down to the same rate as for first order elements (e.g. Thielmann et al., 2014). However, solutions are rarely fully discontinuous and instead contain a mix of smooth and non-smooth regions. Additionally, despite showing the same convergence rate, higher-order elements have still delivered higher accuracy in absolute terms than lower-order methods in many benchmarks (Kronbichler et al., 2012; Thieulot and Bangerth, 2022). We speculate that the same results would be seen for higher-order advection methods in time, although the construction of appropriate benchmarks would be challenging."

**MAIN COMMENT #2:**

**Section 6**

The comparison here is interesting but this section is a bit weak in my opinion.

- Is there any random noise on the initial marker position? Is it exactly the same initial condition for both models?

There is random noise on the marker positions, but we have taken care to make sure the noise is identical between the models (our random number algorithm is reproducible and uses the same seed number for both models). Initial conditions are exactly the same. We have clarified this now in the text by adding the following sentence:

"We solve this model using the particle-based RK2 and RK2FOT advection schemes, and take care to make sure all other parts of the model and algorithms are identical and deterministic except for the advection scheme. In particular, we generate particles in identical and deterministic random locations, we enforce the same minimum (40) and maximum (250) number of particles per cell and make sure all algorithms for particle addition and deletion are deterministic."

- The RF2FOT is first order but it should also converge to a physical solution if the time step is refined. Ideally, the same pattern as for RK2, right?

This is correct, we would expect RK2FOT to converge to the RK2 result at sufficiently high temporal resolution. We have added the following sentence to the discussion in this section:

"Considering our benchmark results of the previous sections, we would expect the RK2FOT method to converge to the same solution as RK2, though requiring substantially smaller time steps; at least in this application it would clearly require time steps that make the model prohibitively expensive."

- What is missing here is a temporal resolution test for the 2 schemes. It would be nice to see whether RF2FOT converges to similar patterns as RK2 if the time step is refined. In fact it's not even clear if the RK2 is close to any physical solution here. The addition of a time convergence study would greatly improve the content of this study. Moreover, if the 2 schemes converge to the same physical solution, you could clearly evaluate the computing effort needed to reach the

"converged" solution pattern with both methods. This could clearly show the benefit of RK2 in a practical case.

This is a great suggestion and we have added results of models with different timesteps to this section. We have modified the paragraph describing the results as follows:

"As can be seen in the bottom panels of Fig. 5, the two advection methods produce noticeably different locations of onset of convection. While the model with a full RK2 advection scheme develops small-scale convection beneath the oceanic plate at a distance of approximately 1900 km from the ridge (from 1893.2 km to 1969.2 km), the model with RK2FOT develops the onset at varying distances from from 1656.0 km to 1916.0 km. These numbers correspond to plate ages of 37.9-39.4 Myr (RK2) and 34.3-38.3 Myr (RK2FOT), respectively. Because of the strong nonlinearity of these models we do not observe a simple convergence to one solution as in the benchmark results for either of the models. However, it is apparent that the RK2 method produces a much more stable onset location of small-scale convection, and a greater similarity of the other downwellings that develop beyond the initial onset. In contrast, the onset of convection varies significantly in the RK2FOT method depending on the time step size. In addition, the downwellings show very different convection structures. One could speculate that the RK2FOT method starts to converge towards the RK2 results for a CFL number of 0.075, however this is not certain given the strong variations in the RK2FOT results."

Ideally, an application that includes material jumps would have been ideal (e.g. D", magma body, lithosphere stratification) but I realise it might be out of the scope of the current study.

We agree that an application with a material jump is an interesting test case for a future study, but we also feel it is beyond the scope of the current study.